# CDKL3 promotes osteosarcoma progression by activating Akt/PKB

Aina He[1,2,*], Lanjing Ma[3,*], Yujing Huang[1,*], Haijiao Zhang[3], Wei Duan[4], Zexu Li[3], Teng Fei[3], Junqing Yuan[5], Hao Wu[6], Liguo Liu[7], Yueqing Bai[5], Wentao Dai[8], Yonggang Wang[1], Hongtao Li[1], Yong Sun[1], Yaling Wang[1], Chunyan Wang[1], Ting Yuan[9], Qingcheng Yang[9], Songhai Tian[2], Min Dong[2], Ren Sheng[3], Dongxi Xiang[10,11,12]

**Osteosarcoma (OS) is a primary malignant bone neoplasm with high frequencies of tumor metastasis and recurrence. Although the Akt/PKB signaling pathway is known to play key roles in tumorigenesis, the roles of cyclin-dependent kinase–like 3 (CDKL3) in OS progression remain largely elusive. We have demonstrated the high expression levels of CDKL3 in OS human specimens and comprehensively investigated the role of CDKL3 in promoting OS progression both in vitro and in vivo. We found that CDKL3 regulates Akt activation and its downstream effects, including cell growth and autophagy. The up-regulation of CDKL3 in OS specimens appeared to be associated with Akt activation and shorter overall patient survival ($P = 0.003$). Our findings identify CDKL3 as a critical regulator that stimulates OS progression by enhancing Akt activation. CDKL3 represents both a biomarker for OS prognosis, and a potential therapeutic target in precision medicine by targeting CDKL3 to treat Akt hyper-activated OS.**

## Introduction

Osteosarcoma (OS) is the most common primary bone malignancy in children (Kansara, 2014; Reed et al, 2017). Advanced combinational therapies composed of intensive multidrug treatment and surgeries have been applied to treat OS, yet the 5-yr survival rate for patients with metastasis or relapse remains disappointing with a statistic of less than 30% (Kager et al, 2003; Mirabello et al, 2009). This stagnation of clinical consequences highlights the critical need for defining molecular mechanisms underlying OS development and exploring novel targeted biomarkers and therapies.

Cyclin-dependent kinase–like 3 (CDKL3) is a cell division control protein 2–related kinase that belongs to cyclin-dependent protein kinase–like (CDKL) family (Haq et al, 2001; Yee et al, 2003). Unlike CDKs, the functional and molecular understandings of CDKLs are much under-explored. CDKL3 was first identified in 2001 involved in cell proliferation and central nervous system development (Haq et al, 2001; Dubos et al, 2008). Although a few studies have linked CDKL3 with cancers, the evidence is far lacking in solid support of its role and mechanisms underlying cancer progression (Ye et al, 2018; Zhang et al, 2018).

Akt/protein kinase B is a pivotal serine/threonine protein kinase that governs numerous cellular processes (Manning & Toker, 2017). Akt can be activated by various signals through PI3K. Once PI3K converts PI4,5P2 into the secondary messenger PIP3, Akt was recruited to the plasma membrane and sequentially phosphorylated at two sites. After the dual activation, Akt then gains full capability to regulate numerous downstream targets by Ser/Thr phosphorylation, mainly including glycogen synthase kinase 3 (GSK3), Forkhead box O (FoxO) transcription factors and mTORC1 (mTOR complex 1). Therefore, from different aspects, Akt crucially controls cell proliferation, cell metabolism, cell cycle, autophagy and apoptosis. Because of the significance abovementioned, PI3K-Akt over-activation is virtually observed in most types of human malignant tumors (Vivanco & Sawyers, 2002; Saxton & Sabatini, 2017b). Functional mutations of PI3KCA and overexpression of AKT directly promote tumorigenesis and are frequently discovered in human cancer genomic studies (Fruman & Rommel, 2014; Mundi et al, 2016). To this point, myriad antibodies and small molecules targeting the key components of Akt-related pathways have been

[1]Department of Oncology, Shanghai Jiaotong University Affiliated Sixth People's Hospital, Shanghai, PR China    [2]Department of Urology, Boston Children's Hospital, Harvard Medical School, Boston, MA, USA    [3]College of Life and Health Sciences, Northeastern University, Shenyang, PR China    [4]School of Medicine and Centre for Molecular and Medical Research, Deakin University, Waurn Ponds, Victoria, Australia    [5]Department of Pathology, Shanghai Jiaotong University Affiliated Sixth People's Hospital, Shanghai, PR China    [6]Department of Vascular Biology, Boston Children's Hospital, Boston, MA, USA    [7]Department of General Surgery, Xinhua Hospital Affiliated to Shanghai Jiao Tong University School of Medicine, Shanghai, China    [8]Shanghai Center for Bioinformation Technology and Shanghai Engineering Research Center of Pharmaceutical Translation, Shanghai Industrial Technology Institute, Shanghai, PR China    [9]Department of Orthopedics, Shanghai Jiaotong University Affiliated Sixth People's Hospital, Shanghai, PR China    [10]Division of Genetics, Department of Medicine, Brigham and Women's Hospital, Boston, MA, USA    [11]Department of Medicine, Harvard Medical School, Boston, MA, USA    [12]Shanghai Research Center of Biliary Tract Disease Affiliated to Shanghai Jiao Tong University School of Medicine, Shanghai, China

Correspondence: anna_1188@126.com; shengren1211@126.com; dxiang@bwh.harvard.edu
*Aina He, Lanjing Ma, and Yujing Huang contributed equally to this work

selected in clinical trials or approved for targeted cancer therapy (Fruman & Rommel, 2014; Mundi et al, 2016).

In this work, we identified the function of CDKL3 in promoting OS progression by using multiple experimental models, including cells, animals, and clinical samples. We deeply investigated the relevant molecular mechanisms and showed the pivotal roles of CDKL3 in Akt regulation. These findings provide CDKL3 as a novel biomarker for examining OS prognosis and may represent a new candidate and prospect on the targeted therapy for Akt hyper-activated malignant tumors.

# Results

### CDKL3 promotes OS cell growth

To study the function of CDKL family kinases in OS, we first collected primary OS tumors and adjacent non-tumor tissues from a few OS patients and performed RT-quantitative PCR (qRT-PCR) assay to detect the expression level of CDKL1-5. Among all CDKLs, CDKL3 and CDKL4 showed significantly enhanced expression in tumor samples compared with adjacent non-tumor tissues (Figs 1A and S1). To consolidate the functional roles of CDKLs in OS, we thus knocked down CDKL1-5 and CDK6 (as a positive control) in human OS cell U2OS by using siRNAs (Table S1 and Fig S2A and B). Silencing of CDK6 and CDKL3 significantly inhibited the growth of U2OS cells compared with the control, with inhibitory rates of 30.49% ± 3.59% and 32.17% ± 3.61%, respectively (P < 0.001), whereas interfering expression of any other CDKLs did not affect the growth of U2OS cells (Fig 1B). The anti-proliferative effect of CDKL3 knockdown was validated by a different approach using shRNA interference (Fig S2B–E). The knockdown of CDKL3 by transforming with lentivirus stably expressing CDKL3-shRNA

(Table S1 and Fig S2B) in U2OS and Saos-2 OS cells (Fig S2C) remarkably reduced the cell growth (Fig 1C) and decreased their colony formation by 36.25% ± 2.16% and 12.82% ± 3.11%, respectively (P < 0.01, Fig 1D and E). These data suggest that CDKL3 promotes OS cell growth in vitro.

### CDKL3 promotes OS cell invasion and migration

To further disclose the role of CDKL3 in OS progression, we developed two stable CDKL3-KO cell lines from U2OS and Saos-2 cells using the CRISPR/Cas9 gene-editing strategy (Ran et al, 2013). Both of Sanger sequencing and Western blot assays verified the successful KO of CDKL3 protein from individual clones (U19 and S4) (Figs 2A and B, S3, and Table S2). Consistent with the inhibition of OS cell growth upon CDKL3 knockdown (Fig 1C), the growth of both CDKL3-KO U2OS and Saos-2 cells was prohibited by GFP-expressed U2OS and Saos-2 cells when cocultured them together in a growth competition assay (Fig S4). These data indicated that overexpression of CDKL3 was able to promote OS cell proliferation. We next explored the role of CDKL3 in OS cell invasion and migration. In the wound healing assay, a wound of ~535-µm (U2OS) and 875-µm (Saos-2) width was generated, and the wound closure was photographed at 0-, 4-, 6-, 20-, 24-, and 48-h post-initial wound generation (Fig 2C–H). Based on the width of the wound, we calculated the speed of cell migration of U19 and S4 cells at 4.99 and 8.17 µm/h, respectively (Fig 2E and H); they were both slower than that of parental and CDKL3-rescued cells (~1.5–2 fold). As shown in Fig S5, the morphology of migrating U2OS and Saos-2 cells with lamellipodia (red circle) was clearly observed.

We next measured invasion and migration of U2OS and Saos-2 cells upon CDKL3-KO using chamber Transwells and found in both cases that their invasiveness was significantly decreased and the

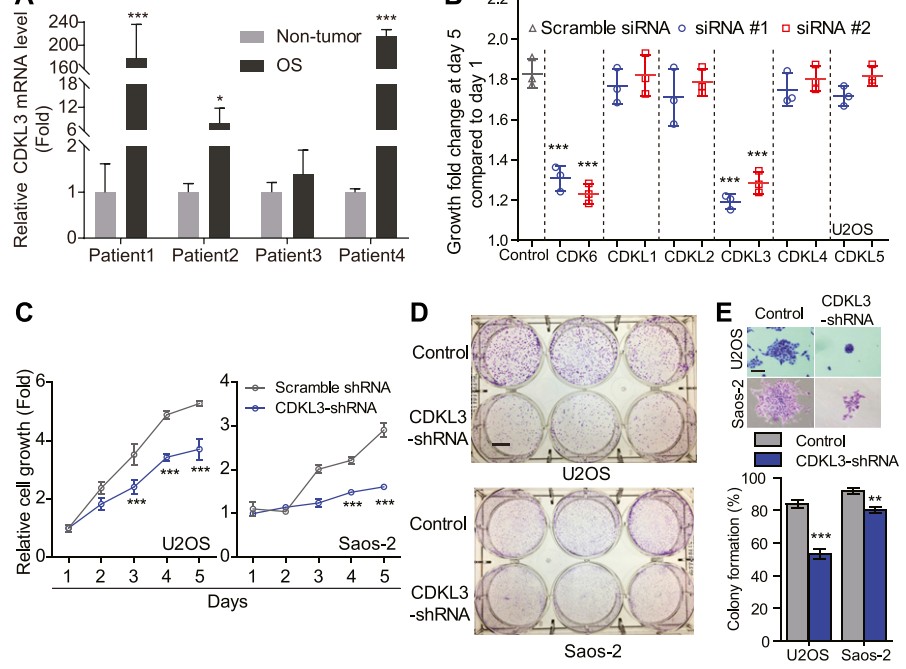

**Figure 1. Cyclin-dependent kinase–like 3 (CDKL3) promotes the growth of osteosarcoma (OS) cells.**
**(A)** Comparison of CDKL3 expression levels by qRT-PCR analysis between adjacent non-tumor tissues and OS tissues derived from four OS patients. Three OS and three non-OS tissues from each patient were collected and analyzed. Two housekeeping genes (ACTB and GAPDH) were included in this study to normalize the CDKL3 expression. **(B)** Cell growth analysis of U2OS cells transfected with indicated siRNAs. **(C)** Cell growth analysis of U2OS and Saos-2 cells with CDKL3-shRNA silencing (n = 5). **(D)** Colony formation assay of cells conducted in (C). Scale bar = 1 cm. **(E)** Representative colony images and quantitative analysis from (D) (n = 3). Scale bar = 200 µm. Error bars represent SD (n = 3). *P < 0.05, **P < 0.01, ***P < 0.001, two-tailed t test.

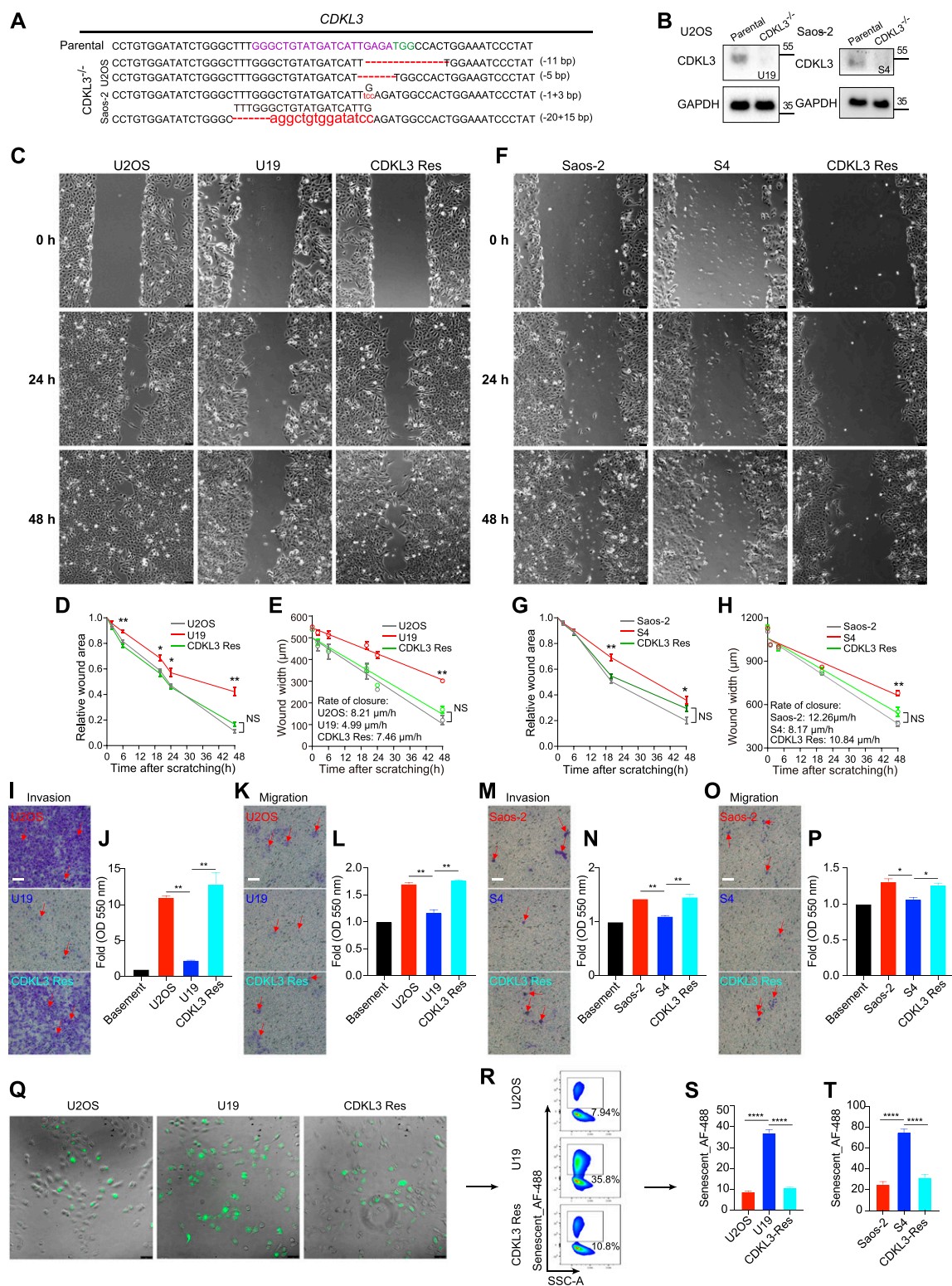

**Figure 2. Knockdown of cyclin-dependent kinase–like 3 (CDKL3) inhibited osteosarcoma (OS) cell invasion and migration.**
**(A)** Genomic DNA sequencing of CDKL3-KO clones. **(B)** CDKL3 expression in CDKL3-KO and parental cells by Western blot analysis. **(C)** Microscopy images of wound closure of parental, CDKL3-KO (U19), and CDKL3-rescued (Res) U2OS cells at 0, 24, and 48 h after scratching. Scale bar = 75 $\mu$m. **(D)** Quantification of the wounded area invaded during 48 h of U2OS cells (n = 3). **(E)** Quantification of wound healing speed ($\mu$m/h) of U2OS cells (n = 3). **(F)** Microscopy images of wound closure of parental, CDKL3-KO (S4), and CDKL3-rescued (Res) Saos-2 cells at 0, 24, and 48 h after scratching. Scale bar = 75 $\mu$m. **(G)** Quantification of the wounded area invaded during 48 h of Saos-2 cells (n = 3). **(H)** Quantification of wound healing speed ($\mu$m/h) of Saos-2 cells (n = 3). **(I)** Representative images of Transwell-invaded U2OS cells (purple stained, red arrows).

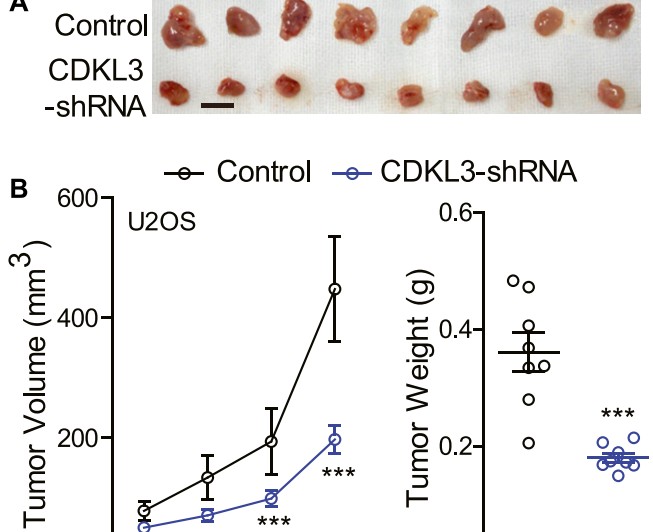

**Figure 3. Knockdown of CDKL3 suppresses OS tumor growth in vivo.**
**(A)** Representative images of tumor xenografts obtained from the indicated groups. Scale bar = 1 cm. **(B)** Left: tumor volumes over treatment. Right: tumor weight at harvest. Error bars represent SD (n = 8). *P < 0.05, ***P < 0.001, two-tailed *t* test.

ability of invasion and migration was converted by rescuing CDKL3 (Fig 2I–P). The ability of cell invasion and migration was quantified by measuring the extraction solution derived from invaded or migrated cells with a plate reader. As shown in Fig 2J, L, N, and P, both of the parental U2OS and Saos-2 cells maintained significant higher percentage of cells that underwent invasion and migration than cells with CDKL3-KO. Cellular senescence, a highly stable state of cell cycle arrest, was believed to limit cancer cell proliferation (Herranz & Gil, 2018; Gorgoulis et al, 2019). Even the role of senescence in cancer progression seems contradicting: either tumor-suppressing or tumor-promoting in a stage-dependent manner, cellular senescence is a popular strategy in cancer treatment for a cell to avoid malignant transformation (Park et al, 2000; Suzuki et al, 2001; Han et al, 2002; te Poele et al, 2002; Herranz & Gil, 2018; Gorgoulis et al, 2019). We tested the cellular senescent possibility of OS cells upon CDKL3-KO (U19 and S4), along with their parental and CDKL3-rescued cells (Murcia et al, 2019). We observed that U19 and S4 cells exhibited more percentages of senescent cells (~4–5-fold higher) than the percentages of their parental cells, which was reverted in cells after rescuing CDKL3 (Fig 2Q–T). These data suggest that CDKL3 may present a negative relation with the OS cell senescence, which may facilitate OS cells to maintain in a malignant state.

## CDKL3 promotes OS progression in vivo

Given that CDKL3 could affect the OS cell proliferation, invasion, and migration in vitro, we next determined how CDKL3 modulated the growth of OS cells in vivo. We s.c. transplanted U2OS cells transfected with lentivirus that was constructed with either sh-CDKL3 or sh-control to athymic nude mice. Tumor size was measured every 4 d starting from its touchable until day 16. As shown in Fig 3A, tumors generated from U2OS cells with CDKL3 knockdown was ~2-fold smaller than that of the control group, along with the significant difference of tumor weight between them (0.180 ± 0.008*g* versus 0.361 ± 0.033*g*, ***P < 0.001) (Fig 3B). These data suggested that consistent with the in vitro readout, CDKL3 might be able to promote OS tumor growth in vivo.

## CDKL3 modulates autophagy in OS

Based on the functional studies that CDKL3 promotes OS progression, we then questioned the underlying mechanisms. We analyzed one OS clinical dataset (GSE21257) for extracting the differential gene expression pattern based on the CDKL3 expression level and performed KEGG signaling pathway enrichment analysis (false discovery rate cutoff < 0.05). Within the list, two closely related pathways that critically govern cell growth and autophagy, PI3K-Akt and AMPK, were highly enriched at the top (Fig 4A–C and Table S3) (Luo et al, 2003; Kim et al, 2011; Laplante & Sabatini, 2012;

Saxton & Sabatini, 2017b; Herzig & Shaw, 2018). Perturbation of PI3K-Akt and AMPK pathway was supposed to regulate autophagy at both early and late stages (Saxton & Sabatini, 2017b; Herzig & Shaw, 2018). Because autophagy was reported to suppress OS progression (Meschini et al, 2008; Yang et al, 2013; Miao et al, 2015), we asked whether CDKL3 was able to modulate autophagy in OS. Autophagy promotion can be reflected either from the emerging fusion between autophagic vacuoles and endolysosomal compartment or improvement of the lysosomal function. To study the autophagosome maturation upon CDKL3 depletion, we stably expressed a tandem fluorescent LC3 (tfLC3) construct composed of eGFP, mRFP, and LC3 to U2OS and Saos-2 cells with and without CDKL3-KO (Kimura et al, 2007). The engineered cells were subjected to serum starvation which is commonly used for inducing autophagy. The early autophagosomes appeared yellow (eGFP and mRFP merging), whereas late autophagic vacuoles exhibited only red fluorescence because of eGFP quenching at acidic pH with autolysosome (Kimura et al, 2007). After starvation of OS cell lines for 6 h, depletion of CDKL3 resulted in a profound enrichment of autophagy maturation as demonstrated by an increase in both autophagosomes and autolysosomes (Fig 4D and E). The autophagy maturation was compensated by restoring CDKL3 protein via transient

Scale bar = 100 μm. **(J)** Quantification of stained/invaded cells in (I) (n = 3). **(K)** Representative images of Transwell-migrated U2OS cells (purple stained, red arrows). Scale bar = 100 μm. **(L)** Quantification of stained/migrated cells in (K) (n = 3). **(M)** Representative images of Transwell-invaded Saos-2 cells (purple stained, red arrows). Scale bar = 100 μm. **(N)** Quantification of stained/invaded cells in (M) (n = 3). **(O)** Representative images of Transwell-migrated Saos-2 cells (purple stained, red arrows). Scale bar = 100 μm. **(P)** Quantification of stained/migrated cells in (O) (n = 3). **(Q)** Representative images of senescent cells stained with CellEvent senescence green probe. Scale bar = 75 μm. **(R)** FACS analysis of green-stained senescent cells in (Q). **(S)** Quantification of senescent U2OS cells in (R) (n = 3). **(T)** Quantification of senescent Saos-2 cells (n = 3). Error bars indicate SD (n = 3). *P < 0.05, **P < 0.01, ***P < 0.001. Statistical significance of the differences was estimated by unpaired two-tailed *t* test. Linear regression was run on the wound width data using the GraphPad Prism software.
Source data are available for this figure.

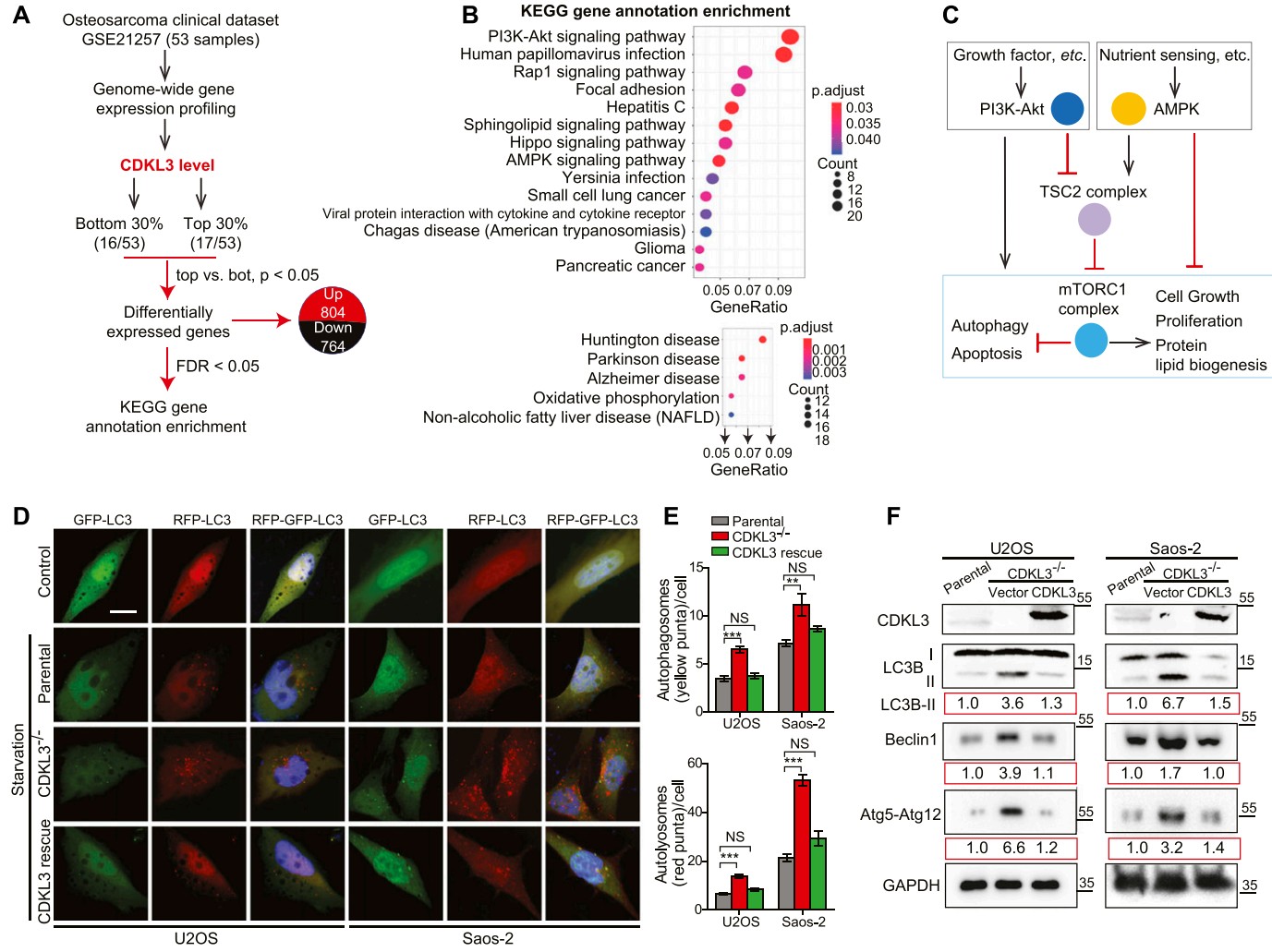

**Figure 4. Cyclin-dependent kinase–like 3 (CDKL3) inhibits autophagy in osteosarcoma (OS).**
**(A)** An outline of OS clinical dataset analysis (GEO accession: GSE21257). 33 out of 53 patient samples were selectively analyzed based on the CDKL3 expression level for the differential gene expression patterns. **(B)** Signaling pathway was mapped by the R package clusterProfiler. Upper and bottom figures represent down-regulated and up-regulated pathways (CDKL3 top 0.3 versus bottom 0.3), respectively. KEGG enrichment analysis using the clusterProfiler R package was performed on differential expressed genes with a strict cutoff of $P < 0.01$ and false discovery rate of less than 0.05. The size of the dot indicates the number of differentially expressed genes in the pathway. Color of the dot represents the value of Benjamini and Hochberg false discovery rate–adjusted $P$-value. **(C)** Molecular connection of PI3K-Akt and AMPK pathway in regulation of cell growth, autophagy, etc. This summarized mechanism was extracted from the GEO dataset GSE21257. mTORC1 plays key roles in eukaryotic cell metabolism by promoting cell growth and inhibiting autophagy and apoptosis. **(D)** Representative images of parental cells and CDKL3-KO U2OS and Saos-2 cells transfected with mRFP-eGFP-LC3, at normal conditions (DMEM with 10% FBS, control) or 6 h EBSS treatment (starvation). Colocalized mRFP and eGFP (yellow dots) indicate autophagosomes, only mRFP (red dots) indicates autolysosomes. Nuclei (blue) were labeled with DAPI. Scale bar = 5 $\mu$m. **(E)** Quantification of autophagosomes (eGFP$^+$mRFP$^+$; yellow) and autolysosomes (eGFP$^-$mRFP$^+$; red) per cell from (D) (n = 3). **(F)** Western blot analysis of LC3B, Beclin-1, and Atg5-Atg12 in indicated cells. The number below each blotting strip is intensity quantification derived from ImageJ. Error bars indicate SEM (n = 3). *$P < 0.05$, **$P < 0.01$, ***$P < 0.001$, two-tailed $t$ test.
Source data are available for this figure.

transfection, which was further consolidated by detection of LC3B, Beclin-1, and Atg12 using Western blot (Fig 4F). On the other hand, apoptosis was not clearly seen in CDKL3 KO cells by both FACS and Western blot analysis (Fig S6). These data together suggested that CDKL3 could inhibit autophagy in OS.

## CDKL3 critically regulates mTORC1 activation in OS

After knowing the autophagy-regulating function of CDKL3, we sought to further investigate the possible upstream mechanisms.

mTORC1 was reported to play central roles in eukaryotic cell metabolism controlled by both PI3K-Akt and AMPK pathway (Laplante & Sabatini, 2012) (Fig 4C). Activation of mTORC1 promotes cell growth, whereas it inhibits autophagy and apoptosis (Laplante & Sabatini, 2012; Saxton & Sabatini, 2017a). To dissect this issue, we used serum starvation assays on parental and CDKL3-KO U2OS cells. Under starvation, parental U2OS cells showed clear reduction in mTOR (S2448), p70-S6K (T389), GSK3 (S9), 4E-BP1, and Akt (both on T308 and S473) phosphorylation, which depicted that the Akt/mTOR signaling cascade was strongly abrogated under such energy/growth

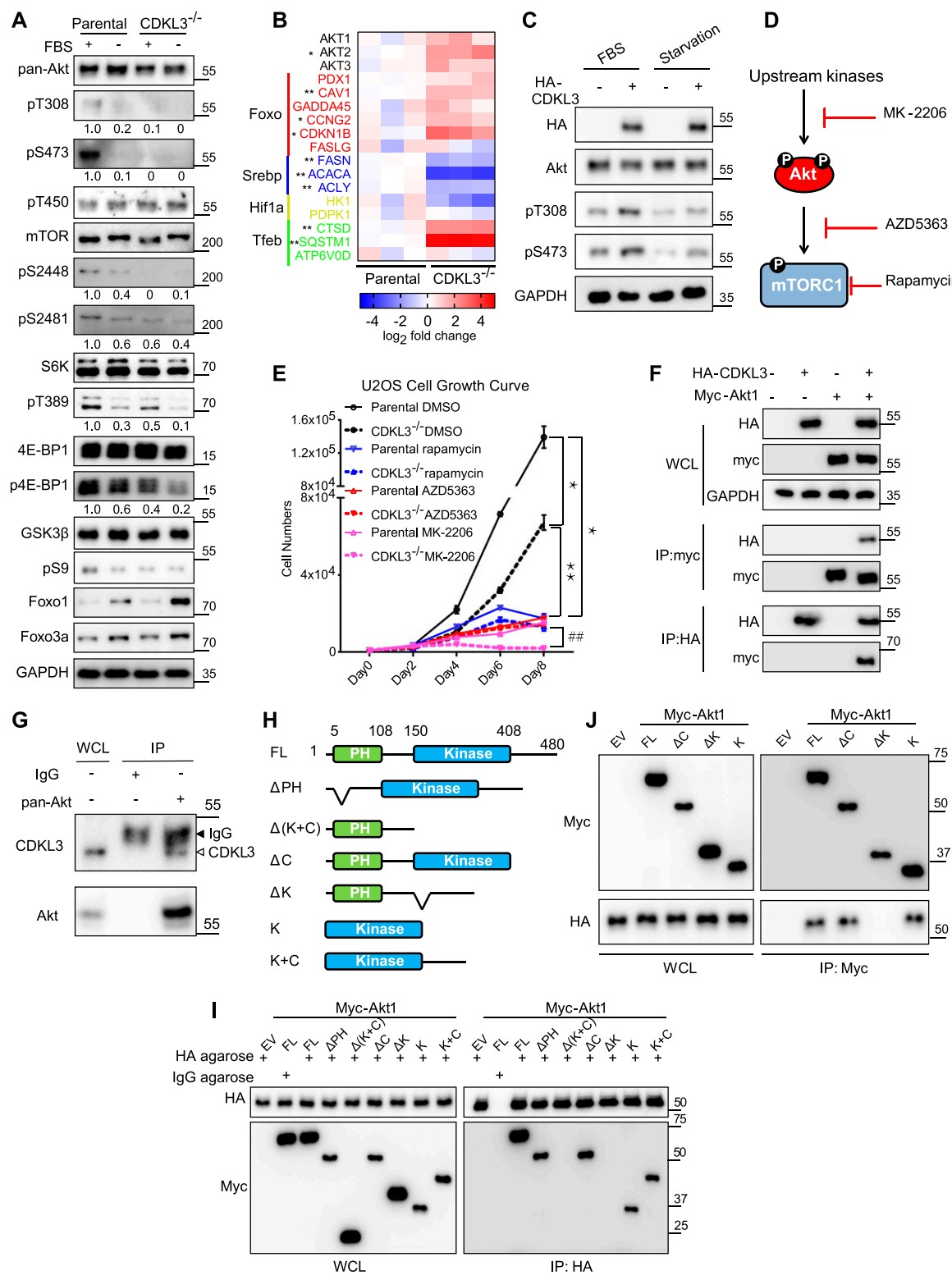

**Figure 5.  Cyclin-dependent kinase–like 3 (CDKL3) critically regulates Akt activation in osteosarcoma (OS).**
**(A)** Akt and mTOR activation under regular or starvation conditions in parental and CDKL3-KO U2OS cells. Upon CDKL3 KO, Akt and mTORC1 activation were significantly alleviated. The numbers below each blotting strip is intensity quantification by ImageJ. **(B)** The expression patterns of mTORC1 and FoxO downstream target genes confirm that CDKL3 regulates both pathways at the transcription level. Cells were cultured under the regular growth condition. **(C)** Overexpression of CDKL3 in U2OS cells causes hyper-activation of Akt in the presence and absence of FBS. **(D)** General working mechanism of rapamycin, AZD5363, and MK-2206. **(E)** U2OS cell growth under different conditions (n = 3). **(F)** Co-immunoprecipitation (Co-IP) reveals the physical interaction between HA-CDKL3 and Myc-Akt1.

factor deprivation circumstance (Sekulic et al, 2000) (Fig 5A). However, in CDKL3-KO cells, significantly diminished phosphorylation of Akt, mTORC1, or their down targets were clearly seen even in the presence of serum, which showed that mTORC1 activation in OS was clearly dependent on CDKL3 levels. At the transcription level, despite of being unexclusive, previous reports have shown that multiple transcription factors such as HIF1α, STAT3, SREBP, and TFEB can be closely regulated by mTORC1 in cancers (Laplante & Sabatini, 2013). Our qRT-PCR data revealed that most of the target genes of abovementioned transcription factors showed significant expression difference comparing parental with CDKL3-KO cells cultured under the regular growth condition (Fig 5B), which further substantiated that CDKL3 critically regulates mTORC1 in OS.

### CDKL3 interacts with Akt and regulates Akt phosphorylation

As shown in Fig 5A, loss of CDKL3 caused significant attenuation of Akt phosphorylation; we then hypothesized that CDKL3 might exert direct effects on Akt to affect its downstream targets. To substantiate this notion, we looked into FoxO transcription factors. FoxOs were known to be phosphorylated by Akt, followed by their nuclear removal and degradation (Zhang et al, 2011). As expected, the protein levels of FoxO1 and FoxO3a were both up-regulated under serum starvation and they were further elevated in CDKL3-KO cells (Fig 5A). As transcription factors, FoxOs are known to govern particular gene transcription which involves in cell cycle and cell death (Carter & Brunet, 2007). By qRT-PCR analysis, classical FoxO-targeted genes arouse ~1–5 folds in the absence of CDKL3, which is consistent with the increased FoxO protein levels (Fig 5A and B). We next interrogated the effect of CDKL3 overexpression in OS cells. Under the regular growth condition, overexpression of CDKL3 strongly enhanced Akt phosphorylation at both T308 and S473 (Fig 5C). Intriguingly, under the 24-h starvation condition, CDKL3 may also strengthen the Akt phosphorylation compared with empty vector. We further monitored the growth of parental and CDKL3-KO U2OS cells after treatment with classical mTORC1 inhibitor (rapamycin), and two Akt inhibitors (AZD5363 and MK-2206) that are currently administrated in clinical trials. AZD5363 perturbates Akt binding to its substrate and MK-2206 inhibits Akt phosphorylation activation (Hirai et al, 2010; Davies et al, 2012). Application of rapamycin and AZD5363 significantly suppressed parental U2OS cell growth (Figs 5D and E and S7). These two inhibitors that act downstream of Akt activation showed similar constrained effects of cell growth on both parental and CDKL3 KO cells, arguing that CDKL3 should function upstream of mTORC1. On the other hand, MK-2206 treatment kept the cell number almost static in CDKL3-KO cells (Fig 5D and E), which was probably due to the total inactivation of Akt phosphorylation by loss of CDKL3 and usage of MK-2206 combinatorically. Taken together,

we demonstrated the significant role of CDKL3 in regulating Akt activation in OS from different angles.

Based on the findings above, we wondered whether CDKL3 can activate Akt by direct contact. The data from co-immunoprecipitation (Co-IP) of CDKL3 with Akt1 showed that CDKL3 coexisted with Akt1 in IPs both exogenously and endogenously (Figs 5F and G and S8). Based on the Akt1 structure, we constructed a series of Akt1 truncations which consist of various functional domains/regions for mapping (Fig 5H). By using Co-IP, we managed to disclose the fact that CDKL3 specifically binds to the kinase domain of Akt1, as the absence of Akt1 kinase domain in any truncation totally lost the binding capacity (Fig 5I and J). We also found that the kinase domain of CDKL3 mainly contributed to the binding, and this interaction was further supported by molecular modeling (Figs S9 and S10). Because CDKL3 is a serine/threonine protein kinase that is capable to bind to and activate Akt, we wondered whether Akt might be a direct substrate of CDKL3. We constructed GST–Akt fusion proteins and purified CDKL3 from cell lysates to perform in vitro kinase assay to interrogate their possible kinase–substrate relationship. When detected by the specific phosphor-Akt antibodies (pT308, pT450, and pS473), however, no clear phosphorylation was seen (data not shown). Collectively, we have shown that CDKL3 physically interacted with Akt on the kinase domain in a specific manner. However, CDKL3 does not phosphorylate the classical Akt activation sites directly. We hypothesize CDKL3 may phosphorylate other undisclosed sites to transactivate Akt, lead to the "open" conformation of Akt upon binding, or adapt Akt to its activation partners.

### CDKL3 defines poor prognosis in clinic by regulating Akt expression

To further investigate the role of CDKL3 in clinic, our in-house survival analysis for OS patients demonstrated that elevated expression of CDKL3 in OS was associated with shorter overall survival ($P$ = 0.003) (Table 1 and Fig 6A), which sustained the similar trend with metastasis-free survival probability (Fig 6B). To determine the consequence of differential expression of CDKL3 in human OS, 152 primary OS tissues on a tissue microarray were examined by immunohistochemistry (IHC) staining (Fig 6C and D). Representative images regarding CDKL3 expression levels (from negative "−" to high expression "+++") were shown in Fig 6C. MKI67 (Ki67) as a critical protein is highly expressed in active cell cycle stages, assigning as a solid tumor proliferation/growth marker (Scholzen & Gerdes, 2000). The overexpression of Ki67 showed a positive correlation with that of CDKL3 ($P$ < 0.001 versus CDKL3-low expressed samples) (Fig 6E), which determines that the shorter survival of patients with CDKL3 overexpression might at least partially be due to subsequent elevated tumor proliferation. As concluded in Fig 5, CDKL3 is able to activate Akt/mTORC pathway by regulating Akt phosphorylation,

**(G)** Endogenous CDKL3 coexists with endogenous Akt shown by co-IP. **(H)** Schematic diagrams of Akt1 constructs for mapping. **(I)** Co-IP of HA-CDKL3 with different Akt1 constructs shows that the Akt kinase domain is indispensable for CDKL3 interaction. **(I, J)** Reverse IP of CDKL3 and Akt1 confirms the findings in (I). Error bars indicate SD (n = 3). *$P$ < 0.05 parental DMSO versus CDKL3$^{−/−}$ DMSO, or parental rapamycin, or parental AZD5363, or parental MK-2206; **$P$ < 0.01 CDKL3$^{−/−}$ DMSO versus CDKL3$^{−/−}$ rapamycin, or CDKL3$^{−/−}$ AZD5363, or CDKL3$^{−/−}$ MK-2206; ##$P$ < 0.01 parental MK-2206 versus CDKL3$^{−/−}$ MK-2206. Statistical significance of the differences was estimated by unpaired two-tailed $t$ test.
Source data are available for this figure.

**Table 1.  Patient characteristics.**

| Characteristic | Number (%) | CDKL3 (−) | CDKL3 (+) | *P*-value |
|---|---|---|---|---|
| Gender | | | | 0.446 |
| Female | 61 (40.1) | 32 (43.2) | 29 (37.2) | |
| Male | 91 (59.9) | 42 (56.8) | 49 (62.8) | |
| Age/year | | | | 0.888 |
| <18 | 71 (46.7) | 35 (47.3) | 36 (46.2) | |
| ≥18 | 81 (53.3) | 39 (52.7) | 42 (53.8) | |
| Tumor site | | | | 1[a] |
| Extremities | 148 (97.4) | 72 (97.3) | 76 (97.4) | |
| Non-extremities | 4 (2.6) | 2 (2.7) | 2 (2.6) | |
| Enneking's surgical staging | | | | 0.396 |
| I/II | 141 (92.8) | 70 (94.6) | 71 (91.0) | |
| III | 11 (7.2) | 4 (5.4) | 7 (9.0) | |
| Karnofsky performance status score | | | | 0.726[a] |
| ≥80 | 146 (96.1) | 72 (97.3) | 74 (94.9) | |
| ≤70 | 6 (3.9) | 2 (2.7) | 4 (5.1) | |
| Neoadjuvant chemotherapy | | | | 1[b] |
| Yes | 150 (98.7) | 73 (98.6) | 77 (91.0) | |
| No | 2 (1.3) | 1 (1.4) | 1 (9.0) | |
| Pathological subtype | | | | 0.242 |
| Conventional | 136 (89.5) | 64 (86.5) | 72 (92.3) | |
| Telangiectatic | 16 (10.5) | 10 (13.5) | 6 (7.7) | |
| Surgical | | | | 0.173 |
| Amputation | 47 (30.9) | 19 (25.7) | 28 (35.9) | |
| Limb salvage | 105 (69.1) | 55 (74.3) | 50 (64.1) | |
| Local recurrence | | | | 0.851 |
| Yes | 38 (25.0) | 18 (24.3) | 20 (25.6) | |
| No | 114 (75.0) | 56 (75.7) | 58 (74.4) | |
| Metastasis | | | | 0.024 |
| Yes | 94 (61.8) | 39 (52.7) | 55 (70.5) | |
| No | 58 (38.2) | 35 (47.3) | 23 (29.5) | |

Pearson's chi-squared test was used to compare clinical differences between patients with negative or positive of CDKL3.
[a]Continuous correction chi-square test.
[b]Fisher's exact test.

leading to the promotion of cancer cell growth and proliferation. This suspicion was further supported by the data shown in OS tissues with the pairing adjacent non-tumor tissues that CDKL3 negatively regulated the bulk Akt, whereas CDKL3 could positively phosphorylate both Akt-T308 and Akt-S473 positions (Fig 6F and G). The IHC staining of OS patient samples confirmed our previous observation that CDKL3 was capable of inhibiting autophagy (LC3 staining) that might be negatively correlated with Akt phosphorylation (Fig 6F and G). Together with prior literatures that alteration of Akt/mTOR signaling pathway was observed in OS (Bishop & Janeway, 2016; Sayles et al, 2019), our data moved it to a further step, demonstrating that CDKL3 may represent a potential targetable subclass of OS.

# Discussion

In this work, we integrated both bench studies and bedside analyses to establish the oncogenic role of CDKL3 in OS progression. CDKL3 emerged as a positive regulator of Akt signaling pathway, autophagy inactivation, and programs that correlate to tumor progression (Fig 6H). OS obtains profound somatic copy number and significant structural alternations with few recurrent point mutations by now identified in protein-coding genes (Perry et al, 2014; Behjati et al, 2017; Sayles et al, 2019). This raises a big challenge to generate effective approaches for precision medicine. Akt has been reported to be one attractive and quantifiable biomarker for

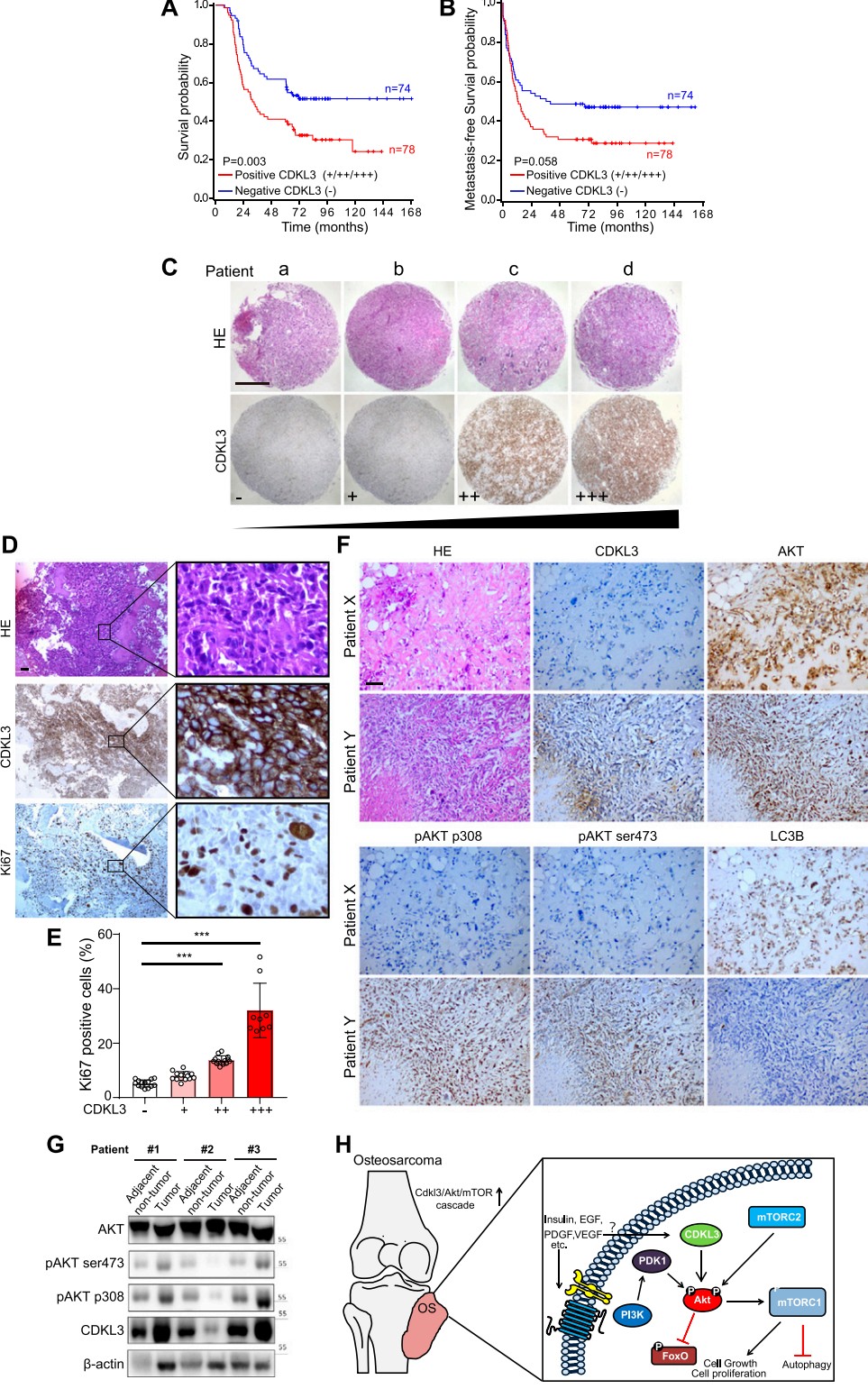

**Figure 6. Cyclin-dependent kinase–like 3 (CDKL3) defines poor prognosis and correlates with Akt phosphorylation in clinic.**
**(A, B)** Kaplan–Meier plots of overall survival (A) and metastasis-free survival (B) of 152 osteosarcoma (OS) patients, stratified by CDKL3 levels (– represents negative staining; +, ++, and +++ represent weak, intermediate, and strong staining, respectively). **(C)** Representative immunohistochemistry (IHC) images of OS biopsies with different levels of CDKL3 expression on an OS microarray containing 152 primary OS tissues samples. Scale bar = 500 μm. **(D)** Representative HE and IHC images of OS biopsies stained by CDKL3 and Ki67. Scale bar = 100 μm. **(E)** Quantitative analysis of Ki67 expression in OS patients with different levels of CDKL3 expression. **(F)** HE and IHC staining images of CDKL3, AKT, pAKT (p308), pAKT (ser473), and LC3 in representative CDKL3-positive and CDKL3-negative patients. Scale bar = 100 μm. **(G)** Western blot detection of CDKL3 and Akt phosphorylation in OS tissues and adjacent non-tumor tissues from three different patients. **(H)** Putative underlying mechanism that CDKL3 promotes OS progression. Overexpression of CDKL3 leads to increased phosphorylation of Akt, followed by governing mTORC1 and FoxO activities, likely independent of functions of PDK1, growth factors, or relevant receptors; this may inhibit autophagy and eventually promote OS development. $*P < 0.05$, $**P < 0.01$, $***P < 0.001$, two-tailed $t$ test. Source data are available for this figure.

predictive and pharmacodynamic purposes (Solomon & Pearson, 2009; Vincent et al, 2011). We present evidence that CDKL3 is capable of directly activating Akt and assign a previously unappreciated function to this serine/threonine kinase. However, we did not observe the direct phosphorylation of Akt by CDKL3 at the traditional sites (T308, T450, and S473) in vitro. This argues that CDKL3 may phosphorylate other sites and contribute to Akt activation, which requires further research by site-specific antibodies or mass

spectrometry, or allosterically activate Akt as suggested in the molecular model, or act as scaffold to bring other kinase for the phosphorylation purpose which was seen in other kinases before (Higuchi et al, 2008; Manning & Toker, 2017). We have unearthed the basic function and mechanism of CDKL3 in OS, but the upstream signal regulating CDKL3 remains obscure. Although early studies showed that CDKL3 activity was insensitive to EGF, our data cannot fully support this notion when considering the PDK1-overactivation effect (Paradis et al, 1999; Collins et al, 2003; Yee et al, 2003). For translational medicine, targeting CDKL3 might be available to provide higher specificity to particular tumor tissues and less toxicity to normal organs. As preclinical and clinical tumor therapies targeting Akt-related pathways spring out robustly in the past decade, we envision that the future CDKL3-specific inhibitors may join the rank either individually or combinatorically (Fruman & Rommel, 2014; Zhang et al, 2015; Mundi et al, 2016).

To conclude, this study integrated both bench and bedside studies to establish the role of CDKL3 in promoting OS progression by potentiating Akt activity and its downstream events. Evaluation of the link between CDKL3 expression and phosphorylation of Akt in OS patients may provide a guide for cancer precision medicine and disease prognosis. Future studies of CDKL3 on different types of tumors and exploring the biological functions of the rest CDKLs in cancer development would be fascinating both in theory and in clinical application.

# Materials and Methods

### Cell culture

Saos-2 and U2OS cells were purchased from the cell bank of Shanghai Biology Institute, Chinese Academy of Science. HEK293T cells were provided by Shanghai Tumor Research Institute. 293T and Saos-2 cells were grown in DMEM medium (Biowest) supplemented with 10% fetal bovine serum (Biowest). U2OS cells were grown in RPMI 1640 medium (Biowest) with 10% fetal bovine serum (Biowest).

### High-content screening for cell growth

Saos-2 cells were seeded at a density of 2,000 cells/well in a 96-well plate, and the cell growth was then monitored once a day for 5 d after shRNA transduction using the Cellomics ArrayScanV high-content image platform and calculated by HCS Studio Cell Analysis Software (Thermo Fisher Scientific).

### cDNA, siRNA, and shRNA constructs

The cDNAs of CDKL3 (HG11563-G; Sino Biological Inc.) was cloned into either pcDNA3.1 vector (V800-20; Invitrogen) or pLenti-Hygro vector (#17484; Addgene). sgRNA-resistant CDKL3 was first cloned into pGEM-T vector using the primer 5′-TGGAAAACCTGTGGA-TATCTGGGCTTTAGGATGCATGATTATCGAAATGGCCACTGGAAATCCCTATCTTCC and amplified using primers 5′-CTGGTTCTGGTTCTGAATTCATGATG-GAGATGTATGAAACCCTTGGAAAAGTGG; 5′-CTCTAGACTCGAGCGGCC-GCTCACCTCACACTGGGGTGAAAGAGATTTGA. The insert with triple-HA tag at their N termini (with GSGSGSEF as linker) was further cloned to

pcDNA3.1 vector via Gibson Assembly (2621; New England Biolabs) and sequenced using primers 5′-GGGCTGATAAAGAAGGCAGAAAATT; 5′-TGCCTTGGCCTCCCAAAGTGCA. Indicated siRNAs (sequences listed in Table S1) were synthesized at Sangon Biotech. siRNAs were transfected reversely in a 96-well plate by using Lipofectamine RNAiMAX Transfection Reagent (#13778030; Thermo Fisher Scientific) according to the manufacturer's protocol. Briefly, RNAiMAX Transfection Reagent and siRNA in Opti-MEM Medium (31985-070; Thermo Fisher Scientific) were incubated for 5 min at RT, followed by adding into cells seeded (5,000 cells/well) in a 96-well plate. The final siRNA concentration applied in the assay was 10 nM. After 12-h incubation, the cells were refreshed by normal culture medium. shRNA (ccgggaGGAGATATCTCAGAACCAActcgagTTGGTTCTGA-GATATCTCCtcttttg) cloned in GV112 plasmid (AgeI; EcoRI; GENCHEM) was delivered into cells by using the lentiviral transfection system. Briefly, the transfection mixture prepared in Lipofectamine TM 2000 (Invitrogen) including 20 µg CDKL3-shRNA plasmid or control plasmid together with 15-µg pHelper 1.0 and 10-µg pHelper 2.0 was added to 293T cells in a 10-cm dish after 15-min incubation at RT. The viral particle supernatant was collected at 48 and 72 h after transfection and filtered through 0.45-µM syringe filters for further experiments. Infected cells were selected with puromycin (2 µg/ml) after 48-h incubation and maintained in puromycin-containing culture medium.

### Bioinformatics analysis

Expression data matrix was obtained from public dataset (GEO accession number GSE21257; 53 OS samples; [Buddingh et al, 2011]) using the GEO query package in R. Differential expression between CDKL3 high expression group (top 30%; 17 samples) and CDKL3 low expression group (bottom 30%; 16 samples) was determined using differential expression analysis for sequence count data (DESeq v1.24.0; [Anders & Huber, 2010]). To obtain more differentially expressed genes rather than several significantly altered genes, a $P$-value cutoff of 0.05 was applied. KEGG gene annotation enrichment analysis using the clusterProfiler R package (Yu et al, 2012) was performed on differential expressed genes with a strict cutoff of $P < 0.01$ and a false discovery rate of less than 0.05. R package enrichplot (https://github.com/GuangchuangYu/enrichplot) achieves several visualization methods to help integrating enrichment results.

### Reverse transcription and quantitative real-time PCR

Total RNAs from cells were purified by either Trizol or the Allprep DNA/RNA mini kit (QIAGEN). cDNA was generated by using iScript (Bio-Rad) according to the manufacturer's protocol. Quantitative RT–PCR (qRT-PCR, for RNA) and PCR (for genomic DNA) were performed using FastStart SYBR Green Master (Roche). All primers are designed based on the primer bank of Massachusetts General Hospital (https://pga.mgh.harvard.edu/cgi-bin/primerbank). The reaction mixtures were incubated at 50°C for 15 min, followed by 95°C for 5 min, and then 35 PCR cycles were performed with the following temperature profiles: 95°C for 15 s, 60°C for 30 s, and 72°C for 1 min. The gene expression values were normalized to those of GAPDH. PCR primers are listed below:

CDKL1-F: ACACGGGTCAGATTGTGGC;
CDKL1-R: GGATTTCCCGAAGGGCAATTT;
CDKL2-F: TCTCCCAGTCTGGCGTTGT;
CDKL2-R: ACCATCGGGTTGCCACATAAT;
CDKL3-F (for siRNA): TTCATCACGAAAACCTGGTCAA;
CDKL3-R (for siRNA): AAGTCGCTTACTCTCTAGTCCAT;
CDKL3-F (for shRNA): TCAAAGGAGGAAGAGGAGA;
CDKL3-R (for shRNA): AGTTAGATTGATGGGTGGC;
CDKL4-F: AGGGTCTTATGGGGTTGTATTCA;
CDKL4-R: AGCTACTACTTGTCCAGAGGTTT;
CDKL5-F: AGGAGCCTATGGAGTTGTACTT;
CDKL5-R: TTTCCTGCTTGAGAGTCCGAA;
CDK6-F: GCTGACCAGCAGTACGAATG;
CDK6-R: GCACACATCAAACAACCTGACC;
GAPDH-F: TGACTTCAACAGCGACACCCA;
GAPDH-R: CACCCTGTTGCTGTAGCCAAA;
ACTB-F: CATGTACGTTGCTATCCAGGC;
ACTB-R: CTCCTTAATGTCACGCACGAT;
AKT1-F: GTCATCGAACGCACCTTCCAT;
AKT1-R: AGCTTCAGGTACTCAAACTCGT;
AKT2-F: GGTGCAGAGATTGTCTCGGC;
AKT2-R: GCCCGGCCATAGTCATTGTC;
AKT3-F: TGAAGTGGCACACACTCTAACT;
AKT3-R: CCGCTCTCTCGACAAATGGA;
CCNG2-F: TCTGTATTAGCCTTGTGCCTTCT;
CCNG2-R: CCTTGAAACGATCCAAACCAAC;
CDKN1B-F: AACGTGCGAGTGTCTAACGG;
CDKN1B-R: CCCTCTAGGGGTTTGTGATTCT;
GADD45A-F: CCCTGATCCCAGGCGTTTTG;
GADD45A-R: GATCCATGTAGCGACTTTCCC;
FASLG-F: CTCCGAGAGTCTACCAGCCA;
FASLG-R: TGGACTTGCCTGTTAAATGGG;
PDX1-F: GGAGCAGGATTGTGCCGTAA;
PDX1-R: CTGTGGGGACGCACTAAGG;
CAV1-F: CATCCCGATGGCACTCATCTG;
CAV1-R: TGCACTGAATCTCAATCAGGAAG;
FASN-F: CCGAGACACTCGTGGGCTA;
FASN-R: CTTCAGCAGGACATTGATGCC;
ACLY-F: ATCGGTTCAAGTATGCTCGGG;
ACLY-R: GACCAAGTTTTCCACGACGTT;
ACACA-F: TCACACCTGAAGACCTTAAAGCC;
ACACA-R: AGCCCACACTGCTTGTACTG;
PDPK1-F: CCTAAGGCAAGAGACCTCGTG;
PDPK1-R: CGTCGTCTTCCGACATAGCC;
HK1-F: CACATGGAGTCCGAGGTTTATG;
HK1-R: CGTGAATCCCACAGGTAACTTC;
SQSTM1-F: GACTACGACTTGTGTAGCGTC;
SQSTM1-R: AGTGTCCGTGTTTCACCTTCC;
CTSD-F: ATTCAGGGCGAGTACATGATCC;
CTSD-R: CGACACCTTGAGCGTGTAG;
ATP6V0D2-F: TCTGATCGAAACGCCATTAGC;
ATP6V0D2-R: CTTCTTTGCTCAATTCAGTGCC.

## Mixed and single clone KO cells

The selected sgRNA sequence (GGGCTGTATGATCATTGAGA) was cloned into lentiGuide-Puro vectors (#52963; Addgene). U2OS and

Saos-2 cells that stably express Cas9 were generated using the Lenti-Cas9-Blast construct (#52962; Addgene) and were selected using blasticidin S (10 µg/ml, B12150.01; RPI). These Cas9-expressing cells were transduced with lentivirus that expresses sgRNA. The mixed stable cell lines were selected using puromycin. Single clones of U2OS or Saos-2 KO cells were generated by diluting the mixed KO cells at about 0.8 cell per well in 48-well plates. The genotype of single-cell clones was determined by amplifying the DNA fragments containing the sgRNA targeting region using the following primers: CDKL3 forward: CTGATCTGTTTCAGACCTGTG; CDKL3 reverse: 5′-GCATCAACCATCAACATATGG, followed by ligating the PCR product into T-vectors (A3600; Promega). The ligation products were transformed into *Escherichia coli* (DH5α strain) and plated onto agar plates. Single-clone colonies were selected, and their plasmids were extracted and sequenced.

## Serum starvation assay

Cells were plated in six-well plates for 80% confluency before use. When the assay began, all medium was removed, and the cells were washed by cold PBS twice. Either DMEM+10%FBS or DMEM alone were added to the cells for 24 h. The cultured medium was removed after reaching designed time courses. The cells were washed by cold PBS twice and subjected to lysis.

## Cell growth assay

U2OS-GFP and Saos-2-GFP stable cell lines were established by transduction of ZsGreen (a gift from Bryan Welm and Zena Werb, plasmid #18121; Addgene) expression lentivirus (Welm et al, 2008), monoclonal cell population was isolated by limiting dilution in 48-well plates. U2OS-GFP expression cells were seeded at 3,000 cells/well in 96-well black-walled plates (Cat. no. 3603; Corning), cell growth was measured daily using FLUOstar Omega Filter–based multimode microplate reader (BMG Labtech) with the excitation wavelength at 485 nm and emission at 510 nm. Growth fold at day N was determined by Day N absorbance divided by Day 1.

## Methyl thiazolyl tetrazolium (MTT) assay for cell proliferation

Cells transduced with lentivirus expressing indicated shRNAs were plated on 96-well plates at a density of 3,000 cells per well in triplicates and incubated for 1–5 d. MTT (thiazolyl blue tetrazolium, from Sigma-Aldrich) was added into each well for a final concentration of 0.5 mg/ml, and the plates were incubated in 37°C for additional 4 h. After the incubation, all the medium was removed and 100 µl of DMSO was added to each well. Then, the test-ready plate was assayed by Synergy H1 microplate reader at $OD_{490}$. Growth curve was drawn by according to relative cell growth fold, which was determined by Day N absorbance divided by Day 1.

## Colony formation assay

Saos-2 and U2OS cells seeded in six-well plates at 800 and 5,000 cells per well in six-well plates on the third day after infection were cultured for 14 d to form colonies. The cells were subsequently treated by washing with PBS, fixing in 4% paraformaldehyde for 15

min, staining with 0.5% crystal violet for 1 h, washing three times by ddH$_2$O and photographing with a digital camera. The number of colonies was counted. All assays were performed in triplicate.

## Cell culture wound healing assay

U2OS, Saos-2 cells and their CDKL3-KO (U19 and S4) cells at a density of ~2 × 10$^5$ were plated in a 12-well plate for 90–100% confluence in 24 h. A straight wound was created inside a biosafety hood by manually scraping the cell monolayer with a 1000-$\mu$l pipette tip followed by aspirating the medium and cell debris with PBS, and the full culture medium was then carefully added against the well wall without detaching cells. Following photographing the initial wounds, the plate was placed back to an incubator at 37°C with 5% CO$_2$. At several time points after scraping (e.g., 0, 4, 6, 20, 24, and 48 h), the snapshot pictures of the wound closure were taken, and the healing area was measured under the microscope (Leica). At least six different wound areas were photographed for each cell types.

## Cell invasion and migration assay

The cell invasion and migration assay was conducted according to the manufacturer's introduction (CBA-100-C; Cell Biolabs) (Airoldi et al, 2016; Li et al, 2017). Briefly, 300 $\mu$l of warm serum-free medium was added to the cell culture inserts to rehydrate the basement membrane for 1 h at RT, 500 $\mu$l of complete cell culture medium supplemented with 10% FBS was added to the lower well of the invasion or migration plate. 300 $\mu$l of cell suspension in the serum-free medium (1.5 × 10$^5$ cells) was seeded to the inside of each insert and incubated for 24 h in a cell culture incubator at 37°C. The medium inside the insert was carefully aspirated and any noninvasive cells were removed using 2–3 cotton swabs. The insert was then stained with 400 $\mu$l of cell stain solution, incubated for 10 min at RT, and washed three times with water. The dried insert membrane was ready for imaging under a microscope. To quantitively analyze the degree of invasion and migration of indicated cells, 200 $\mu$l of extraction solution was added to each membrane in an empty well, followed by incubating for 10 min on an orbital shaker. 100 $\mu$l of the extraction solution was transferred to a 96-well microtiter plate and the OD at 550 nm was measured in a plate reader. CellEvent Senescence Green Detection Kit from Invitrogen (C10850) was used to detect senescence-associated $\beta$-galactosidase (SA-$\beta$-gal) activity following the manufacturer's instructions (Murcia et al, 2019).

## Growth of cells in athymic nude mouse and tumor size determination

U2OS cells transfected with Lv-sh Control or Lv-sh CDKL3 were trypsinized, washed, and resuspended in PBS. Designed cell number and viability were determined using trypan blue. Sixteen 6–8-wk-old male athymic nude mice (*BALB/cASlac-nu, from Shanghai SLAC laboratory animal CO. LTD*) were randomly divided into two groups (8 mice/group) followed by s.c. injecting with U2OS cells at a density of 1 × 10$^6$ cells per site. The tumor size was determined every 3–4 d after the tumor was formed as previously described (Xiang et al, 2017). Tumor volumes were measured, and the mice were weighed twice weekly. Tumor photographing and weighting were conducted on day 16

immediately after euthanizing tumor-bearing mice. Tumor volume was calculated using the formula: ½ (Length × Width$^2$). All animal experiments were approved by the Animal Care and Use Committee of Shanghai Sixth People's Hospital (Approval No.: 2016-0107).

## Growth competition assay

Wild-type GFP-expressing U2OS and Saos-2 cells were cocultured with CDKL3-KO U2OS and Saos-2 cells at different ratios, and the cells were passaged when reaching 80–90% confluence (as indicated at days post coculture in Fig S6), the percentage of GFP-positive cells was measured by FACS.

## Autophagy induction

CDKL3-KO and wild-type U2OS and Saos-2 cells were cultured in eight-well chamber slides (#154534; Thermo Fisher Scientific) to 70–80% confluence, and the cells were then co-transfected with a tandem fluorescent-tagged RFP-GFP-LC3 reporter plasmid (ptfLC3, #21074; Addgene) and sgRNA-resistant CDKL3 or control plasmid using PolyJet (#SL100688; SignaGen Laboratories) according to the manufacturer's protocol. The medium was changed into culture medium 24 h post-transfection, and the cells were subjected to 6-h starvation with EBSS medium (E2888; Sigma-Aldrich) on the next day. The cells were then fixed in 4% paraformaldehyde and imaged using the Olympus DSU-IX81 Spinning Disk Confocal System. Images were pseudocolored and analyzed using ImageJ. Yellow dots (eGFP$^+$ mRFP$^+$) were considered as autophagosomes, whereas RFP only (red dots) as autolysosomes.

## Small molecule growth inhibition assay and IC50 assay

For the small molecule inhibitors, rapamycin, AZD5363, and MK-2206 were all purchased from Selleckchem and dissolved in DMSO. In the growth assay, the cells were plated into 24-well plates at 1,000 cells to start with for each well. After plating for 4 h, "day 0" wells were trypsinized and counted by a cell counter manually. Each condition was plated into three wells and counted three times. After that, the cells were counted by the same approach every 48 h till "day 8." The concentrations of rapamycin, AZD5363, and MK-2206 used in the assay are 2 nM, 80, and 15 $\mu$M, respectively. Equal amount of DMSO was added for the control groups. For the IC$_{50}$ assay, 2,000–3,000 cells were plated into each well of 96-well plate. Different concentrations of small molecules were added into each well by the same volume, as DMSO was added to the control group. After 72 h, the MTT assay will be tested as indicated above.

## In vitro kinase assay

The GST-Akt was cloned into pGEX4T-1 vector for *E. coli* expression. The *E. coli* strand *BL21* was used for plasmid transformation. After transformation, a single colony was inoculated into LB medium for growth with ampicillin at 37°C. At OD$_{600}$ = 0.6, 0.2 mM IPTG was added to the bacterial culture for overnight induction at 18°. Next day, the cell pellets were harvested by centrifugation and lysed in buffer (50 mM Tris–HCl, 200 mM NaCl, and 10% [vol/vol] glycerol, pH = 7.9) by sonication on ice. After sonication, the lysate was

centrifuged, and the supernatant was subjected to GST-resin (GenScript) for incubation at 4°C for 2 h. The GST-resin was then applied to the gravity column. After reduced glutathione elution, the proteins went through the size exclusion column (Akta) for purification. HA-CDKL3 was transfected into HEK293T cell lines in a 10-cm dish. After transfection by Lipofectamine 2000 (Invitrogen) for 24 h, the petri dish was refreshed by clean medium. After 48 h, the cells were harvested and lysed. The HA-conjugated agarose was added to the whole cell lysate to enrich HA-CDKL3. After enrichment, HA epitope peptide (YPYDVPDYA, synthesized by Biotherm, purity >95%) was used to elute HA-CDKL3 from resin. All proteins were checked by Western blot before in vitro kinase assay. For the in vitro kinase assay, proper amount of GST-resin (GST alone or GST-Akt) with HA-CDKL3 or BSA was mixed together in 1× Kinase Buffer (Hepes 20 mM, MgCl$_2$ 50 mM, sodium orthovanadate 1 mM, EDTA 1 mM, and DTT 1 mM, pH = 7.5). At the condition of adding ATP (from Santa Cruz Biotech, final concentration in reaction: 0.2 mM) or distilled water, the reactions were performed at 30° for 30 min. Then, the reactions were stopped immediately by adding 3× SDS-containing Laemmli buffer at due time.

## Western blot and Co-IP

The antibodies were purchased from the following companies: Cell Signaling Technology: LC3, Pan-Akt (Rb and Mo), Akt pT308, Akt pT450, GAPDH, p70S6K, p70-S6K pT389, mTOR, GSK3$\beta$, GSK3$\beta$ pS9, 4E-BP1, p4E-BP1, Atg12, Beclin-1, HA, myc, GST, and IgG; Invitrogen: CDKL3 (NKIAMRE) and Akt pS473; ImmunoWay: mTOR pS2448 and mTOR pS2481; ProteinTech: FoxO1 and FoxO3a; and Santa Cruz Biotech: $\beta$-actin and protein G agarose. The HA antibody–conjugated and myc antibody–conjugated agarose was purchased from Pierce. For Western blot, the cell was lysed by Passive Lysis Buffer (25 mM Tris–HCl, 150 mM NaCl, and 1% NP40) containing protease inhibitor cocktail (Roche). Total protein amount was determined by the BCA assay (Pierce), and appropriate amount of denatured protein with Laemmli loading dye was loaded onto 6% or 10% poly-acrylamide gel for SDS–PAGE. After that, the PVDF membrane was used for transfer, and TBS-T buffer containing 5% BSA with 0.02% sodium azide was used in membrane blocking and antibody incubations. The chemiluminescent substrate kit was purchased from Tanon. Final quantification of gel intensity was performed by ImageJ and plotted in Prism 8.0. For co-IP, the total lysate of cell was subjected to $\alpha$-HA-agarose, $\alpha$-myc-agarose, or incubated with $\alpha$-Akt/IgG and protein G agarose at 4°C overnight. Next day, the resin was washed thoroughly by TBS-T for five times with 10-min incubation and shaking intervals at 4°C before subjected to SDS–PAGE and Western blot.

## Tissue microarray

A total of 152 formalin-fixed, paraffin-embedded OS tissue samples were randomly selected for tissue microarray construction. Areas-of-interest tumor regions were identified and marked on the source block. Representative 1-mm diameter tissue core from each case were punched from the original block and inserted into a new recipient paraffin block using a tissue arraying instrument (Beecher

Instruments), and 4-$\mu$m sections were cut using a microtome and mounted on a glass slide.

## IHC analysis

Paraffin sections (4-$\mu$m thickness) were deparaffinized and treated with 3% hydrogen peroxide for 10 min to quench the endogenous peroxidase activity. Antigenic retrieval was performed by submerging in citric acid (pH 6.0) and microwaving. The slides were then allowed to cool at RT, followed by incubation in normal goat serum for 1 h to block nonspecific binding, then incubated overnight at 4°C using the following primary antibodies: CDKL3 (NKIAMRE, PA5-72930; Invitrogen), Ki67 (clone SP6, ab16667; Abcam), AKT Pan Monoclonal antibody (clone J.314.4, MA5-14916; Invitrogen), phospho-AKT1 (Thr308, clone B18HCLC, 710122; Invitrogen), Phospho-Akt (Ser473, clone D9E, 4060; Cell Signaling Technology), and LC3B (2775; Cell Signaling Technology). The staining was examined using HRP EnVision Systems (Dako). Rabbit IgG was used as a primary antibody for negative control. Each section was evaluated by three independent pathologists without the knowledge of the clinical case features. The results of IHC staining were evaluated independently by two trained pathologists without the knowledge of clinical data. The specimens stained for CDKL3 and NKIAMRE were graded into three groups according to the strength of the staining color: – represents no staining and +, ++, and +++ represent weak, intermediate, and strong staining, respectively. For Ki-67 staining, a labeling index was assessed by counting the percentage of positively stained cells in different regions containing 500 cells in each core.

## Molecular docking procedure

Docking experiments were performed using ClusPro 2.0 program (Comeau et al, 2004a, 2004b). The 3D crystal structure of human CDKL3 (PDB ID: 3ZDU) and Akt1 (PDB ID: 6BUU) were obtained from the Protein Data Bank. The Akt1 monomer was docked into the CDKL3 to generate predicted binding models of CDKL3:Akt1. Models with the highest scores and good topologies were selected for the proposed models of the interaction between CDKL3 and Akt1. Images were generated using PyMoL.

## Cases

152 OS patients admitted to Shanghai Sixth People's Hospital from 2006 to 2014 were enrolled in this study. Pathological results were based on specimens obtained at diagnosis. All the patients received surgery and adjuvant chemotherapy; the surgical margins were classified according to the Enneking staging system. Patients were followed at a 3-mo interval and their clinical data were updated for at least 5 yr. The patients' clinical characteristics including age, gender, pathologic subtypes, tumor site, Enneking surgical stage, Karnofsky Performance status score, recurrence, and metastases were collected for statistical analysis. Progression-free survival and overall survival were calculated from the date of initial diagnosis. Consent for collection of patient specimens and using patient characteristics for this study and publication was approved by the Independent Ethics Committee of the Shanghai Sixth

People's Hospital (Approval No.: 2018-079). All patients signed informed consent forms.

## Statistical analysis

All statistical analyses were performed using the SPSS statistical software (version 19.0; IBM Corp.) or Prism8 (GraphPad). Overall survival was defined as the time from the date of diagnosis to the date of the last follow-up or death from any cause. Chi-square test or Fisher's exact test was used to compare clinical differences between patients with negative or positive of CDKL3. Kaplan–Meier analysis was performed to generate survival curves. An unpaired two-tailed $t$ test was used for comparisons between two experimental groups. $P$-values were two-sided, and a $P < 0.05$ was considered statistically significant.

# Supplementary Information

# Acknowledgements

This study was supported by Natural Science Foundation of Shanghai, China (16ZR1425900), and St. Baldrick's Foundation (585350), the Fundamental Research Funds for the Central Universities, China (N182005006), the Starting Funds for "100-talent" plan of Northeastern University, China (01270021920501), National Natural Science Foundation of China (31970721 and 81672736), and National Key R&D Program of China (2018YFC0910500).

## Author Contributions

A He: conceptualization, resources, data curation, formal analysis, and methodology.
L Ma: data curation and formal analysis.
Y Huang: data curation and formal analysis.
H Zhang: resources, data curation, and formal analysis.
W Duan: validation, investigation, and methodology.
Z Li: resources, data curation, and formal analysis.
T Fei: resources, data curation, software, and formal analysis.
J Yuan: resources, data curation, and formal analysis.
H Wu: resources, software, and formal analysis.
L Liu: resources, data curation, and formal analysis.
Y Bai: resources, data curation, software, and formal analysis.
W Dai: resources, data curation, software, and formal analysis.
Y Wang: resources, data curation, and formal analysis.
H Li: resources, data curation, and formal analysis.
Y Sun: resources, data curation, and formal analysis.
Y Wang: resources, data curation, and formal analysis.
C Wang: resources, data curation, and formal analysis.
T Yuan: resources, data curation, and formal analysis.
Q Yang: resources, data curation, and formal analysis.
S Tian: resources, data curation, and validation.
M Dong: resources, validation, and visualization.
R Sheng: conceptualization, resources, data curation, software, formal analysis, validation, and investigation.
D Xiang: conceptualization, resources, formal analysis, validation, investigation, and visualization.

## Conflict of Interest Statement

The authors declare that they have no conflict of interest.

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
