## [Reviewer comments · Life Science Alliance]

Life Science Alliance

CDKL3 promotes osteosarcoma progression by activating Akt/PKB

Aina He, Lanjing Ma, Yujing Huang, Haijiao Zhang, Wei Duan, Zexu Li, Teng Fei, Junqing Yuan, Hao Wu, Liguang Liu, Yueqing Bai, Wentao Dai, Yonggang Wang, Hongtao Li, Yong Sun, Yaling Wang, Chunyan Wang, Ting Yuan, Qingcheng Yang, Songhai Tian, Min Dong, Ren Sheng, and Dongxi Xiang

DOI: <https://doi.org/10.26508/lsa.202000648>

Corresponding author(s): Dongxi Xiang, Brigham and Women's Hospital/Harvard Medical School; Aina He, Shanghai Jiaotong University Affiliated Sixth People's Hospital; and Ren Sheng, Northeastern University

Review Timeline:

Submission Date:	2020-01-13
Editorial Decision:	2020-02-03
Revision Received:	2020-03-03
Editorial Decision:	2020-03-14
Revision Received:	2020-03-18
Accepted:	2020-03-19

Scientific Editor: Andrea Leibfried

Transaction Report:

February 3, 2020

Re: Life Science Alliance manuscript #LSA-2020-00648-T

Dr. Dongxi Xiang
Brigham and Women's Hospital/Harvard Medical School
Medicine
Boston, MA 02115

Dear Dr. Xiang,

Thank you for submitting your manuscript entitled "CDKL3 promotes osteosarcoma progression by activating Akt/PKB" to Life Science Alliance. The manuscript was assessed by expert reviewers, whose comments are appended to this letter.

As you will see, the reviewers have different opinions on the value provided to others by your findings. Reviewer #2, who is positive on your work, confirmed during our cross-commenting session that the concerns put forward by reviewer #1 are valid. The main concern of this reviewer is the preliminary nature of some of your findings and the low number of replicates as well as the limited evidence provided for the conclusions put forward.

We discussed your work in light of this input within our editorial team. We would like to invite you to submit a revised version of your manuscript to us, addressing the specific concerns of both reviewers. We realize that the concern pertaining to the preliminary status of the work cannot get fully addressed, but we would like to encourage you to provide at least some more support for an oncogenic role of CDKL3 in osteosarcoma progression as well as for the proposed link to autophagy in order to provide a significant value to others.

Thank you for this interesting contribution to Life Science Alliance. We are looking forward to receiving your revised manuscript.

Sincerely,

B. MANUSCRIPT ORGANIZATION AND FORMATTING:

Reviewer #1 (Comments to the Authors (Required)):

The authors aim to investigate the oncogenic role of CDKL3 in osteosarcomas. If this protein represents a new possible target in OS, the study could bring interesting new insights. The major points are 1) CDKL3 controls OS cell proliferation but not apoptosis 2) CDKL3 control AKT activation.

However, the results are very confusing. The authors should group all the data showing the link between CDKL3 and proliferation in vitro, in vivo and in patients instead of scattering them in each paragraph.

Showing that CDKL3 interacts with and controls AKT may be significant but the downstream effects on AKT effectors such as mTOR are then obvious. As shown, it is expected that suppressing AKT/mTOR activation abrogated the inhibition of autophagy induction in serum-starved cells. Conversely CDKL3 overexpression does not suppress autophagy in OS cells. So it seems difficult to conclude that CDKL3 control autophagy.

It is not exact to claim that it is well documented that autophagy suppress OS progression. At the opposite, many results indicated that autophagy is involved in chemoresistance and is not a death process but a mechanism that may lead to cell-self-sufficiency. Autophagy protects cancer cells from senescence.

It is not clear whether the effect on CDKL3 on AKT is independent of PI3K as suggested in the recapitulative figure4G.

Other major points:

In almost figures, the number of biological replicates is missing. EX: Fig1A The RT-PCR analysis of CDKLs expression in each patient. Are the results means of technical replicates from one RNA extract from one patient or means of values obtained from different extracts from one patient? Why has this analysis been performed only in 4 patients? The results show "significant" increase of CDKL3 only in 2/4. The results should be presented as % of mean of 2 housekeeping genes to compare the level of CDK3 expression in the different patients as well as in tumor/normal tissues. The photos in fig4C,D are not informative or may be not clearly explained. In how many positive and negative tumors was performed the IHC analysis presented in fig4F?

There are too many speculative conclusions.

In conclusion, this is a very preliminary work that cannot lead to establish the oncogenic role of CDKL3 in OS progression, mainly because OS progression does not only depend on cell proliferation. Moreover, the role of AKT in cancer cells is already strongly documented. The work must further explore the role of CDK3 in cell migration and metastatic potential. This work is not ready for publication.

Reviewer #2 (Comments to the Authors (Required)):

In the present study, the authors investigated the contribution of CDKL3 to the progression of osteosarcoma through the Akt/PKB pathway. Overall, the experiments are well conducted and the manuscript is clear. There are some minor points to be addressed as listed below.

1. (Abstract) "biosmarker" might be a typo. Should it be "biomarker" ?
2. (Figure 1F, G) Author should indicate when the photo was taken and the tumor weight was measured (what day after xenograft). A scale bar should be shown in F.
3. (Figure 2C) Authors should explain more in the legend. Is this based on the present data or just a schema for known knowledge?
4. (Figure 2F-H) Information about the rescue experiment with CDKL3 overexpression looks lacking.

5. Figure 3B Authors should indicate the medium condition, starved or regular?
6. Figure 3 Information about the error bar and p-values is lacking in the legend.
7. (Page 6, line15) "wide" should be "wild".

Dear Editors,

We highly appreciate the detailed consideration of our manuscript provided by you and Reviewers, including the supportive comments. We have performed additional experiments or rearranged the data presentation in our attempt to respond scientifically to the Reviewer's comments. The revised manuscript contains **25** new figure panels. We find the manuscript much improved and we again thank you and Reviewers for the time and expert commentary.

In the following point-by-point response, we have *italicized* each Reviewer point. Any significant changes made to the manuscript, including text (marked in blue) and figures are extracted for Reviewer's review. We also extensively polished the manuscript; any minor amendment of the text will not be listed here but are marked in blue in the revised manuscript.

Reviewer #1 (Comments to the Authors (Required)):

The authors aim to investigate the oncogenic role of CDKL3 in osteosarcomas. If this protein represents a new possible target in OS, the study could bring interesting new insights. The major points are 1) CDKL3 controls OS cell proliferation but not apoptosis 2) CDKL3 control AKT activation. However, the results are very confusing.

We thank the Reviewer for their time and careful consideration of our study.

The authors should group all the data showing the link between CDKL3 and proliferation in vitro, in vivo and in patients instead of scattering them in each paragraph.

We agree with the Reviewer's comment that it might be better to present the data following the orders as *in vitro*, *in vivo* and *in patients*. In the revised manuscript we now isolated the *in vivo* data (Figure 1F-G) and make it as a new individual Figure 3. We also moved and integrated the patient data (Figure 3F) to the new Figure 6G panel.

The logical strategy of data presentation is now updated as below:

- In vitro** ← **Figure 1** The role of CDKL3 in OS proliferation *in vitro*
 We still maintained the Figure 1A patient data but with updated exhibition pattern, this is an important indication that why it is worth to study the role of CDKL3 in OS.
- In vivo** ← **Figure 2** The role of CDKL3 in OS cell invasion and migration *in vitro* (new data required)
- In vivo** ← **Figure 3** The role of CDKL3 in controlling OS progression *in vivo*
- Figure 4** Introduction of the correlation of CDKL3 expression with PI3K/Akt signaling, which was supported by assessing the role of CDKL3 in regulating autophagy
 We'd like to emphasize here that studying the role of CDKL3 in regulation of autophagy is not the main point of this manuscript, investigating this part is to validate the matters of studying CDKL3 in regulating Akt signaling pathway.
- Figure 5** Comprehensive biochemical and molecular mechanisms regarding the role of CDKL3 in regulating Akt and Akt phosphorylation
- In patient** ← **Figure 6** *In patient* data validating the *in vitro* and *in vivo* studies, and providing a conclusive/mechanistic diagram

Below, we have extracted the significant changes that were updated in the revised manuscript.

Results Page 7:

CDKL3 promotes OS progression *in vivo*

Given that CDKL3 could affect the OS cell proliferation, invasion and migration *in vitro*, we next determined how CDKL3 modulated the growth of OS cells *in vivo*. We subcutaneously (*s.c.*) transplanted U2OS cells transfected with lentivirus that was constructed with either sh-CDKL3 or sh-control to athymic nude mice. Tumor size was measured every four days starting from its touchable until the day 16. As shown in Figure 3A, tumors generated from U2OS cells with CDKL3 knockdown was ~2-fold smaller than that of the control group, along with the significant difference of tumor weight between them ($0.180 \pm 0.008\text{g}$ vs. $0.361 \pm 0.033\text{g}$, $***P < 0.001$) (Figure 3B). These data suggested that consistent with the *in vitro* readout, CDKL3 might be able to promote OS tumor growth *in vivo*.

New Figure 3

Page 27 (Figure Legend):

Figure 3. Knockdown of CDKL3 suppresses OS tumor growth *in vivo*. (A) Representative images of tumor xenografts obtained from the indicated groups. (B) Left: Tumor volumes over treatment. Right: Tumor weight at harvest. Scale bar = 1 cm. Error bars represent SD (n=8). * $P < 0.05$, *** $P < 0.001$.

Showing that CDKL3 interacts with and controls AKT may be significant but the downstream effects on AKT effectors such as mTOR are then obvious. As shown, it is expected that suppressing AKT/mTOR activation abrogated the inhibition of autophagy induction in serum-starved cells. Conversely CDKL3 overexpression does not suppress autophagy in OS cells. So it seems difficult to conclude that CDKL3 control autophagy.

We thank for the reviewer's insightful question and would respectfully explain our strategy again. In order to establish the connection between CDKL3 and autophagy in OS, we employed CRISPR-Cas9 system to KO CDKL3 in two OS cell lines as shown in Figure 2A-B. In Figure 4D-F, we have shown that CDKL3-loss significantly induced autophagy, as rescuing by exogenous CDKL3 reverted the phenomenon. We used this KO and rescue system because of its physiological relevance.

In fact, we did not perform "CDKL3 overexpression" to evaluate the correlation between CDKL3 and autophagy. It is due to a simple strategy that CDKL3 already expresses at moderate level in the OS cells, and if CDKL3 was overexpressed we envisioned a good chance that CDKL3 might exceed certain level and would exert no further effect. For example, there are two major possibilities. One, CDKL3 may function together or through other proteins, which is so far unknown. Thus, solely expression of CDKL3 may reach certain threshold and provides no additional suppression of autophagy. Two, as revealed in our later manuscript, CDKL3 regulates Akt and we argued that CDKL3 modulates autophagy through Akt-mTOR. It was well-documented and reviewed a decade ago that a negative-feedback loop is implanted in Akt signaling to prevent its hyper-activation (Manning et al., *Genes Dev*, 2005; Hay, *Cancer Cell*, 2005). Accordingly, it is possible that in these OS cells even though Akt showed stronger phosphorylation, it does not necessarily conclude the further alleviation of autophagy through mTOR.

Taken together, we believe that by employing KO/rescue system we have already step up a strong basis of CDKL3 regulation of autophagy. And we'd also like to kindly point out that as depicted in the title and abstract, the major focus of this manuscript is on CDKL3 regulation of Akt.

It is not exact to claim that it is well documented that autophagy suppress OS progression. At the opposite, many results indicated that autophagy is involved in chemoresistance and is not a death process but a mechanism that may lead to cell-self-sufficiency. Autophagy protects cancer cells from senescence.

The Reviewer raises an important question. Autophagy indeed has been reported to be involved in chemoresistance (e.g., by enhancing cell stress tolerance) (Pan et al., *Mol. Cancer*, 2014; Su et al.; *PLoS ONE*, 2016). However, the role of autophagy in cancer progression remains contradicting: either tumor-promoting or tumor-suppressing in a stage-dependent manner (Cristofani et al., *Cell Death Dis*, 2018; Rao et al., *Nat. Communications*, 2014; Koustas et al., *Am J Clin Oncol*, 2019; Gao et al. *Cell Death Dis*, 2019). Similarly, autophagy was originally found to suppress cellular senescence by removing damaged cellular components, yet more and more studies also suggested that autophagy promoted cellular senescence (e.g., by aiding the synthesis of senescence-associated secretory proteins) (Dou et al., *Nature*, 2015; Dou et al., *Autophagy*, 2016; Horikawa et al., *Nat. Communication*, 2014, Nam et al., *Autophagy*, 2013; Amaravadi et al., *J. Clin. Invest.*, 2007). These pro-senescence and anti-senescence dual roles of autophagy may reflect a complex picture of autophagic regulation on cellular senescence. The key point of this manuscript is to investigate the relation between CDKL3 with Akt signaling, the autophagic reaction upon differential CDKL3 expression included in our study was to support the report that high expression of CDKL3 was able to perturbate PI3K-Akt pathway since the PI3K-Akt pathway was supposed to regulate autophagy (Herzig and Shaw, 2018; Saxton and Sabatini, 2017a). Our data shown that CDKL3 could inhibit autophagy in OS cells (Figure 4). We did not want to focus too much on the mechanisms that how CDKL3 regulates autophagy in this study. We also have not supposed to illustrate how in detail the autophagy process regulates OS. Our new data indicated that the CDKL3 expression level seems possessing a negative correlation with the cellular senescence (Figure 2Q-T), which may facilitate OS cells to maintain in a malignant state. All the detailed mechanisms regarding how CDKL3 regulates autophagy and cellular senescence, and how autophagy regulates OS progression, are subjected to further investigation in our following projects.

Below, regarding this comment we amended relevant text in the revised manuscript to make them become more accurate or moderate, one sentence is extracted as below:

Results Page 8:

We amended the sentence “Since autophagy ~~is well documented~~ to suppress OS progression” to “Since autophagy ~~was reported~~ to suppress OS progression”.

Results Page 9:

“The data together suggested that CDKL3 ~~strongly inhibited~~ autophagy in OS.” was amended to “These data together suggested that CDKL3 ~~could~~ inhibit autophagy in OS.”

It is not clear whether the effect on CDKL3 on AKT is independent of PI3K as suggested in the recapitulative figure4G.

The Reviewer raises an interesting question. Indeed, we did not have adequate resource at this moment to fully address the question that how CDKL3 is regulated upstream. As a very novel kinase, we would like to first establish its tumor promoting role of CDKL3 and pinpoint its mechanism to Akt-related signaling. Answering the question that whether CDKL3 is controlled by growth factors or PI3K is our ongoing work. We sincerely hope the reviewer may understand our situation and intentions.

We'd like to respectfully let the reviewer understand that addressing the question of whether CDKL3 is dependent on PI3K may be very complicated. For instance, PDK1 is the major kinase that phosphorylates Akt at T308. In normal condition, this process requires PI3K which converts PIP2 into PIP3 to recruit Akt to the proximity of PDK1. However, early reports also shown that over-expression of PDK1 or mutating PDK1 to its constitutively active form may phosphorylate Akt in the absence of serum or growth factor (Maurer et al., *Cancer Res*, 2009; Wick et al., *J Biol Chem*, 2000). It means that under such conditions, PDK1 may phosphorylate Akt independent of PI3K, which is different from normal conditions. In our manuscript, we performed one overexpression assay (Fig 3C). We observed that similar with PDK1, overexpression of CDKL3 enhanced Akt phosphorylation in the presence or absence of serum. However, based on the background we described above; we can only draw the conclusion that CDKL3 indeed regulates Akt phosphorylation instead of its dependence of PI3K.

In fact, there is a long way to go for the understanding of CDKL3 regulation of Akt. First we need to identify whether CDKL3 phosphorylates Akt on unknown sites by Mass-spec as negative results were shown on the traditional sites. And then we will identify its other substrates. Only if one of these questions is addressed first, we can then clearly dissect the upstream of CDKL3. Therefore, we can only finish this work first and pursue the reviewer's question in our future studies.

Other major points:

In almost figures, the number of biological replicates is missing. EX: Fig1A The RT-PCR analysis of CDKLs expression in each patient. Are the results means of technical replicates from one RNA extract from one patient or means of values obtained from different extracts from one patient? Why has this analysis been performed only in 4 patients? The results show "significant" increase of CDKL3 only in 2/4. The results should be presented as % of mean of 2 housekeeping genes to compare the level of CDK3 expression in the different patients as well as in tumor/normal tissues.

We thank the reviewer pointed out this question. We now clarified the number of biological replicates in figure legends accordingly.

As for the detailed patient information of Figure 1A, we indeed included two housekeeping genes (ACTB and GAPDH) in our original data, we thus re-arranged and re-presented the data as requested

by the reviewer. Regarding this comment, in the revised manuscript the resulting images include a new Figure 1A, and a new Supplementary Figure 1.

New Figure 1A

Page 24 (Figure legend):

Figure 1. **CDKL3 promotes the growth of OS cells.** (A) Comparison of CDKL3 expression levels by RT-qPCR analysis between adjacent non-tumor tissues and OS tissues derived from four OS patients. Three OS and three non-OS tissues from each patient were collected and analyzed. Two housekeeping genes (ACTB and GAPDH) were included in this study to normalize the CDKL3 expression....

New Supplementary Figure 1

Supplementary Page 16:

Supplementary Figure 1. **CDKL family kinase expression levels in OS patients.** (A) Comparison of CDKL1, 2, 4, 5 expression levels by RT-qPCR analysis between adjacent non-tumor and OS tissues. Relative mRNA levels were calculated using the $2^{-\Delta Ct}$ method, with mean of ACTB and GAPDH as internal controls. CDKL3

expression in OS tissues was normalized to its paired non-tumor tissue from the same patient. (B) CDKL family kinase expression in four OS patients. Data are presented as mean \pm SD. * $P < 0.05$, ** $P < 0.01$, *** $P < 0.001$.

The photos in fig4C,D are not informative or may be not clearly explained. In how many positive and negative tumors was performed the IHC analysis presented in fig4F?

Regarding this point, we extracted the amended text in the revised manuscript as below:

Results Page 12:

To determine the consequence of differential expression of CDKL3 in human OS, one hundred and fifty-two primary OS tissues on a tissue microarray were examined by immunohistochemistry (IHC) staining (Figure 6, C-D). Representative images regarding CDKL3 expression levels (from negative “-” to high expression “+++”) were shown in Figure 6C.

Page 33 (Figure legend):

(C) Representative IHC images of OS biopsies with different levels of CDKL3 expression on an OS microarray containing 152 primary OS tissues samples.

There are too many speculative conclusions.

Thank you for identifying the oversight. To make any conclusions or assumptions be more accurate or moderate, we have modified or removed such text accordingly as extracted below:

Abstract Page 2:

“...and comprehensively investigated the oncogenic role of CDKL3 both *in vitro* and *in vivo*...” was amended to “...and comprehensively investigated the role of CDKL3 in promoting OS progression both *in vitro* and *in vivo*...”

Results Page 4:

“In this work, we identified the oncogenic function of CDKL3 in OS by using multiple experimental models including cells, animals and clinical samples.” was amended to “In this work, we identified the function of CDKL3 in promoting OS progression by using multiple experimental models including cells, animals and clinical samples.”

Discussion Page 14:

“To conclude, this study integrated both bench and bedside studies to establish the ~~oncogenic~~ role of CDKL3 in **promoting** OS progression by potentiating Akt activity and its downstream events.” was amended to “To conclude, this study integrated both bench and bedside studies to establish the role of CDKL3 in **promoting** OS progression by potentiating Akt activity and its downstream events.”

In conclusion, this is a very preliminary work that cannot lead to establish the oncogenic role of CDKL3 in OS progression, mainly because OS progression does not only depend on cell proliferation. Moreover, the role of AKT in cancer cells is already strongly documented. The work must further explore the role of CDK3 in cell migration and metastatic potential. This work is not ready for publication.

To evaluate the role of CDKL3 in cell migration and metastatic potential, we performed new extensive experiments in testing the function of CDKL3 in regulating OS cell wound-healing, cell migration/invasion and cellular senescent. Please see below the extracted main text and figures in the revised manuscript:

Results Page 5-7:

CDKL3 promotes OS cell invasion and migration

To further disclose the role of CDKL3 in OS progression, we developed two stable CDKL3-knockout (CDKL3-KO) cell lines from U2OS and Saos-2 cells using the CRISPR/Cas9 gene-editing strategy (Ran et al., 2013). Both of sanger sequencing and western blot assays verified the successful knockout of CDKL3 protein from individual clones (U19 and S4) (Figure 2, A and B, S3, and Supplementary Table 3). Consistent with the inhibition of OS cell growth upon CDKL3 knockdown (Figure 1C), the growth of both CDKL3-KO U2OS and Saos-2 cells was prohibited by GFP-expressed U2OS and Saos-2 cells when co-cultured them together in a growth competition assay (S4). These data indicated that overexpression of CDKL3 was able to promote OS cell proliferation. We next explored the role of CDK3 in OS cell invasion and migration. In the wound-healing assay, a wound of approximately 535 μm (U2OS) and 875 μm (Saos-2) wide was generated and the wound closure was photographed at 0, 4, 6, 20, 24 and 48-hour (h) post initial wound generation (Figure 2C-H). Based on the width of the wound, we calculated the speed of cell migration of U19 and S4 cells at 4.99 and 8.17 $\mu\text{m}/\text{h}$, respectively (Figure 2E and H); they were both slower than that of parental and CDKL3-rescued cells (~1.5-2 fold). As shown in Supplementary Figure 5, the morphology of migrating U2OS and Saos-2 cells with lamellipodia (red circle) was clearly observed.

Furthermore, we measured invasion and migration of U2OS and Saos-2 cells upon CDKL3-KO using chamber transwells and found in both cases, their invasiveness was significantly decreased, and the ability of invasion

and migration was converted by rescuing CDKL3 (Figure 2I-P). The ability of cell invasion and migration was quantified by measuring the extraction solution derived from invaded or migrated cells with a plate reader. As shown in Figure 2J, L, N and P, both of the parental U2OS and Saos-2 cells maintained significant higher percentage of cells that underwent invasion and migration than cells with CDKL3-KO. Cellular senescence, a highly stable state of cell cycle arrest, was believed to limit cancer cell proliferation (Gorgoulis et al., 2019; Herranz and Gil, 2018). Even the role of senescence in cancer progression seems contradicting: either tumor-suppressing or tumor-promoting in a stage-dependent manner, cellular senescence is a popular strategy in cancer treatment for a cell to avoid malignant transformation (Gorgoulis et al., 2019; Han et al., 2002; Herranz and Gil, 2018; Park et al., 2000; Suzuki et al., 2001; te Poele et al., 2002). We tested the cellular senescent possibility of OS cells upon CDKL3-KO (U19 and S4), along with their parental and CDKL3-rescued cells (Murcia et al., 2019). We observed that U19 and S4 cells exhibited more percentages of senescent cells (~4-5-fold higher) than the percentages of their parental cells, which was reverted in cells after rescuing CDKL3 (Figure 2Q-T). These data suggest that CDKL3 may present a negative relation with the OS cell senescence, which may facilitate OS cells to maintain in a malignant state.

New Figure 2:

Page 25-26 (Figure legend):

Figure 2. Knockdown of CDKL3 inhibited OS cell invasion and migration. (A) Genomic DNA sequencing of CDKL3-KO clones. (B) CDKL3 expression in CDKL3-KO and parental cells by western blot analysis. (C) Microscopy images of wound closure of parental, CDKL3-KO (U19), and CDKL3-Rescued (Res) U2OS cells at 0, 24 and 48h after scratching. Scale bars = 75 μm. (D) Quantification of the wounded area invaded during 48h of U2OS cells. (E) Quantification of wound healing speed (μm/h) of U2OS cells. (F) Microscopy images of wound closure of parental, CDKL3-KO (S4), and CDKL3-Rescued (Res) Saos-2 cells at 0, 24 and 48h after

scratching. Scale bars = 75 μm . (G) Quantification of the wounded area invaded during 48h of Saos-2 cells. (H) Quantification of wound healing speed ($\mu\text{m}/\text{h}$) of Saos-2 cells. (I) Representative images of transwell invaded U2OS cells (purple stained, red arrows). (J) Quantification of stained/invaded cells in (I). (K) Representative images of transwell migrated U2OS cells (purple stained, red arrows). (L) Quantification of stained/migrated cells in (K). (M) Representative images of transwell invaded Saos-2 cells (purple stained, red arrows). (N) Quantification of stained/invaded cells in (M). (O) Representative images of transwell migrated Saos-2 cells (purple stained, red arrows). (P) Quantification of stained/migrated cells in (O). (Q) Representative images of senescent cells stained with CellEvent™ senescence green probe. (R) FACS analysis of green-stained senescent cells in (Q). (S) Quantification of senescent U2OS cells in (R). (T) Quantification of senescent Saos-2 cells. Error bars indicate SD (n=3). * $P < 0.05$, ** $P < 0.01$, *** $P < 0.001$. Statistical significance of the differences was estimated by unpaired two-tailed Student's t-test. Linear regression was run on the wound width data using the GraphPad Prism software.

New Supplementary Methods

Page 9-10:

Cell culture wound-healing assay

U2OS, Saos-2 cells and their CDKL3-KO (U19 and S4) cells at a density of $\sim 2 \times 10^5$ were plated in a 12-well plate for 90-100% confluence in 24 hours. A straight wound was created inside a biosafety hood by manually scraping the cell monolayer with a 1000 μL pipette tip followed by aspirating the media and cell debris with PBS, full culture media was then carefully added against the well wall without detaching cells. Following photographing the initial wounds, the plate was placed back to an incubator at 37°C with 5% CO₂. At several time points after scraping (e.g. 0, 4, 6, 20, 24 and 48 h), the snapshot pictures of the wound closure were taken, and the healing area was measured under the microscope (Leica). At least 6 different wound areas were photographed for each cell types.

Cell invasion and migration assay

The cell invasion and migration assay was conducted according to the manufacturer's introduction (Cell Biolabs, CBA-100-C) (Airoldi et al., 2016; Li et al., 2017). Briefly, 300 μL of warm serum-free media was added to cell culture inserts to rehydrate the basement membrane for 1 h at room temperature, 500 μL of complete cell culture media supplemented with 10% FBS was added to the lower well of the invasion or migration plate. 300 μL of cell suspension in serum free media (1.5×10^5 cells) was seeded to the inside of each insert and incubate for 24 hours in a cell culture incubator at 37°C. The media inside of the insert was carefully aspirated and any non-invasive cells were removed by using 2-3 cotton tipped. The insert was then stained with 400 μL of cell stain solution, incubated for 10 minutes at room temperature, and washed three times with water. The dried

insert membrane was ready for imaging under a microscope. To quantitatively analyze the degree of invasion and migration of indicated cells, 200 μ L of extraction solution was added to each membrane in an empty well, followed by incubating for 10 minutes on an orbital shaker. 100 μ L of the extraction solution was transferred to a 96-well microtiter plate and measure the OD at 550 nm in a plate reader. CellEvent senescence green detection kit from Invitrogen (C10850) was used to detect senescence-associated β -galactosidase (SA- β -gal) activity following the manufacturer's instructions (Murcia et al., 2019).

New Supplementary Figure 5:

Supplementary Page 20 (Figure legend):

Supplementary Figure 5. **Morphology of migrating U2OS and Saos-2 cells derived from the wound-healing assay.** Migration cells (green arrows) show obvious tails lamellipodia (red circle).

Reviewer #2 (Comments to the Authors (Required)):

In the present study, the authors investigated the contribution of CDKL3 to the progression of osteosarcoma through the Akt/PKB pathway. Overall, the experiments are well conducted, and the manuscript is clear. There are some minor points to be addressed as listed below.

We thank the Reviewer for the precious time and careful consideration of our study.

1. (Abstract) "biosmarker" might be a typo. Should it be "biomarker" ?

We have corrected in the revised Abstract.

2. (Figure 1F, G) Author should indicate when the photo was taken and the tumor weight was measured (what day after xenograft). A scale bar should be shown in F.

The detailed information in Figure 1F-G (it now becomes the new Figure 3A-B in the revised manuscript) was added including the missed scale bar in Figure 3A.

Supplementary Methods

Page 10:

Tumor photographing and weighting were conducted on day 16 immediately after sacrificing the mice-bearing tumors.

Page 27 (Figure legend):

Figure 3: ...Scale bar = 1 cm. Error bars represent SD (n=8). * $P < 0.05$, *** $P < 0.001$.

3. (Figure 2C) Authors should explain more in the legend. Is this based on the present data or just a schema for known knowledge?

We have updated the figure legend as suggested. They are extracted below.

Page 29 (Figure legend):

...(C) Molecular connection of PI3K-Akt and AMPK pathway in regulation of cell growth, autophagy *etc.*. This summarized mechanism was extracted from the GEO dataset GSE21257. mTORC1 plays key roles in eukaryotic cell metabolism by promoting cell growth and inhibiting autophagy and apoptosis....

4. (Figure 2F-H) Information about the rescue experiment with CDKL3 overexpression looks lacking.

The detailed information about the rescue experiment was now included.

Results Page 8:

The autophagy maturation was compensated by restoring CDKL3 protein via transient transfection, which was further consolidated by detection of LC3B, Beclin1 and Atg12 using Western Blot (Figure 4F).

Supplementary Page 2-3:

The cDNAs of CDKL3 (HG11563-G, Sino Biological Inc) was cloned into either pcDNA3.1 vector (Invitrogen, V800-20) or pLenti-Hygro vector (Addgene, #17484). SgRNA-resistant CDKL3 was firstly cloned into pGEM-T vector using the primer: 5'-

TGGAAAACCTGTGGATATCTGGGCTTTAGGATGCATGATTATCGAAATGGCCACTGGAAATCCCTA
TCTTCC, and amplified using primers: 5'-

CTGGTTCTGGTTCTGAATTCATGATGGAGATGTATGAAACCCTTGAAAAAGTGG; 5'-

CTCTAGACTCGAGCGGCCGCTCACCTCACACTGGGGTGAAAGAGATTTGA. The insert with triple-HA tag at their N-termini (with GSGSGSEF as linker) was further cloned to pcDNA3.1 vector via Gibson Assembly (NEB, 2621), and sequenced using primers: 5'-GGGCTGATAAAGAAGGCAGAAAATT; 5'-
TGCCTTGGCCTCCCAAAGTGCA...

5. (Figure 3B) Authors should indicate the medium condition, starved or regular?

This an excellent question, the medium condition related to Figure 3B (now refers to Figure 5B) was included, both in the figure legends and in the main body text:

Results Page 9-10:

... most of the target genes of abovementioned transcription factors showed significant expression difference comparing parental with CDKL3-KO cells cultured under the regular growth condition (Figure 5B), ...

Page 31 (Figure legend):

...(B) The expression patterns of mTORC1 and FoxO downstream target genes confirm that CDKL3 regulates both pathways at the transcription level. Cells were cultured under the regular growth condition....

6. (Figure 3) Information about the error bar and p-values is lacking in the legend.

We have updated the legends as suggested. These are extracted below.

Page 31 (Figure legend):

...Error bars indicate SD (n=2). * $P < 0.05$ Parental DMSO vs CDKL3^{-/-} DMSO, or Parental rapamycin, or Parental AZD5363, or Parental MK-2206; ** $P < 0.01$ CDKL3^{-/-} DMSO vs CDKL3^{-/-} rapamycin, or CDKL3^{-/-} AZD5363, or CDKL3^{-/-} MK-2206; ### $P < 0.01$ Parental MK-2206 vs CDKL3^{-/-} MK-2206.

7. (Page 6, line15) "wide" should be "wild".

We thank the Reviewer for noticing this error. This sentence was amended on Page 6 to “...the growth of both CDKL3-KO U2OS and Saos-2 cells was prohibited by GFP-expressed U2OS and Saos-2 cells when co-cultured them together in a growth competition assay (S4).”

March 14, 2020

RE: Life Science Alliance Manuscript #LSA-2020-00648-TR

Dr. Dongxi Xiang
Brigham and Women's Hospital/Harvard Medical School
Medicine
77 Avenue Louise Pasteur
NRB458
Boston, MA 02115

Dear Dr. Xiang,

Thank you for submitting your revised manuscript entitled "CDKL3 promotes osteosarcoma progression by activating Akt/PKB". As you will see, reviewer #2 appreciates the introduced changes. Reviewer #1 refused to re-review your work, and we therefore evaluated your response to this reviewer's concerns independently. While we think that some of the findings indeed remain preliminary, we also appreciate your balanced response to the concerns raised and the additional wound closure and migration assays added. We therefore concluded that the work paves avenues for future research, and that we can move forward towards publication here. Before sending you an official acceptance letter, a few editorial issues still need your attention:

- Please upload all figures, including supplementary figures, as separate files and without figure legend. The figure legends should all be listed in the main manuscript file
- Please make sure to include "figure" when mentioning your supplementary figures in the manuscript text
- Please provide all tables in either word docx or excel format (2 tables are currently provided as pdfs)
- Please add scale bars to Fig. 1D, E, Fig. S2, Fig. S5 and increase those for better visibility in Fig. 2C,F, Q, Fig. 6C,D, F
- Please add the statistical tests used to each figure legend next to the p-value mentioned
- There are still instances where the number of replicates (be it biological or technical) are unclear, please add this information throughout the manuscript
- Please incorporate the supplementary methods into the main manuscript file
- Please have all corresponding authors link their ORCID iD to their profiles in our submission system (they should have received an email with instructions on how to do so)
- Methods section: For the experiments involving human subjects, please identify the committee approving the experiments and include a statement that informed consent was obtained from all subjects and that the experiments conformed to the principles set out in the WMA Declaration of Helsinki and the Department of Health and Human Services Belmont Report. (see also <https://www.life-science-alliance.org/editorial-policies#humans>)
- Methods section: For studies reporting experiments on live vertebrates and/or higher invertebrates, the corresponding author must confirm that all experiments were performed in accordance with relevant guidelines and regulations. The manuscript must include a statement in the Materials and Methods identifying the institutional and/or licensing committee approving the experiments. (see also <https://www.life-science-alliance.org/editorial-policies#animals>)

A. FINAL FILES:

B. MANUSCRIPT ORGANIZATION AND FORMATTING:

Sincerely,

Reviewer #2 (Comments to the Authors (Required)):

This revised version is greatly improved, and the authors are to be commended. I recommend that it be accepted for publication.

Dear Editors,

We very much thank you for the careful consideration of our revised manuscript and conditionally accepting it for publication in *LSA*.

In the following point-by-point response, we maintained each Editor's comment and provided the responsive text accordingly marked in red. In the submission system, a manuscript file with tracked changes enabled was provided in the category of Related Manuscript file; the tracked changes ACCEPTED document was uploaded as the main manuscript for copyediting or publication.

Thank you for submitting your revised manuscript entitled "CDKL3 promotes osteosarcoma progression by activating Akt/PKB". As you will see, reviewer #2 appreciates the introduced changes. Reviewer #1 refused to re-review your work, and we therefore evaluated your response to this reviewer's concerns independently. While we think that some of the findings indeed remain preliminary, we also appreciate your balanced response to the concerns raised and the additional wound closure and migration assays added. We therefore concluded that the work paves avenues for future research, and that we can move forward towards publication here. Before sending you an official acceptance letter, a few editorial issues still need your attention:

- Please upload all figures, including supplementary figures, as separate files and without figure legend. The figure legends should all be listed in the main manuscript file

Response: all figures have been uploaded individually in PDF files.

- Please make sure to include "figure" when mentioning your supplementary figures in the manuscript text

Response: the word "Fig" has been implied when mentioning supplementary figures in the manuscript text.

- Please provide all tables in either word docx or excel format (2 tables are currently provided as pdfs)

Response: the Table 2 has been provided in excel.

- Please add scale bars to Fig. 1D, E, Fig. S2, Fig. S5 and increase those for better visibility in Fig. 2C,F, Q, Fig. 6C,D, F

Response: those figure panels have been amended as required.

- Please add the statistical tests used to each figure legend next to the p-value mentioned

Response: the statistical information has been added in figure legend.

- There are still instances where the number of replicates (be it biological or technical) are unclear, please add this information throughout the manuscript

Response: the information of biological or technical replicates has been included accordingly.

- Please incorporate the supplementary methods into the main manuscript file

Response: the supplementary methods has been incorporated into the main manuscript file.

- Please have all corresponding authors link their ORCID iD to their profiles in our submission system (they should have received an email with instructions on how to do so)

Response: the corresponding authors have been informed to link their ORCID iD to their personal profiles.

- Methods section: For the experiments involving human subjects, please identify the committee approving the experiments and include a statement that informed consent was obtained from all subjects and that the experiments conformed to the principles set out in the WMA Declaration of Helsinki and the Department of Health and Human Services Belmont Report. (see also <https://www.life-science-alliance.org/editorial-policies#humans>)

Response: the indicated information has already been included.

- Methods section: For studies reporting experiments on live vertebrates and/or higher invertebrates, the corresponding author must confirm that all experiments were performed in accordance with relevant guidelines and regulations. The manuscript must include a

statement in the Materials and Methods identifying the institutional and/or licensing committee approving the experiments. (see also <https://www.life-science-alliance.org/editorial-policies#animals>)

Response: no live vertebrates and/or higher invertebrates are applied in this study.

Response: it would be great if the press release date can be made before April 10.

To upload the final version of your manuscript, please log in to your account: <https://lsa.msubmit.net/cgi-bin/main.plex>

A. FINAL FILES:

Response: we uploaded the final file in .DOCX.

Response: High-resolution (supplementary) figures have been provided individually.

Response: the summary blurb has already been included in the Abstract page and also included in the submission system.

B. MANUSCRIPT ORGANIZATION AND FORMATTING:

Response: the original uncropped/-processed electrophoretic blots have been provided in the category of "Source Data" files.

Sincerely,

Andrea Leibfried, PhD
Executive Editor
Life Science Alliance
Meyrhofstr. 1
69117 Heidelberg, Germany
t +49 6221 8891 502
e a.leibfried@life-science-alliance.org
www.life-science-alliance.org

Reviewer #2 (Comments to the Authors (Required)):

This revised version is greatly improved, and the authors are to be commended. I recommend that it be accepted for publication.

March 19, 2020

RE: Life Science Alliance Manuscript #LSA-2020-00648-TRR

Dr. Dongxi Xiang
Brigham and Women's Hospital/Harvard Medical School
Medicine
77 Avenue Louise Pasteur
NRB458
Boston, MA 02115

Dear Dr. Xiang,

Thank you for submitting your Research Article entitled "CDKL3 promotes osteosarcoma progression by activating Akt/PKB". It is a pleasure to let you know that your manuscript is now accepted for publication in Life Science Alliance. Congratulations on this interesting work.

DISTRIBUTION OF MATERIALS:

Again, congratulations on a very nice paper. I hope you found the review process to be constructive and are pleased with how the manuscript was handled editorially. We look forward to future exciting submissions from your lab.

Sincerely,
